# Dynamic bandwidth allocation in time division multiplexed passive optical networks: a dual-standard analysis of ITU-T and IEEE standard algorithms

Kamran Ali Memon[1], Syed Saeed Jaffer[2], Muhammad Ali Qureshi[3] and Khurram Karim Qureshi[1,4]

[1] Interdisciplinary Research Center for Communication Systems and Sensing, King Fahd University of Petroleum and Minerals, Dhahran, Eastern Province, Saudi Arabia
[2] Institute of Industrial Electronics Engineering (IIEE), Karachi, PCSIR, Pakistan
[3] Department of Information & Communication Engineering, The Islamia University of Bahawalpur, Bahawalpur, Punjab, Pakistan
[4] Optical Communications and Sensors Laboratory (OCSL) and Department of Electrical Engineering, King Fahd University of Petroleum and Minerals, Dhahran, Saudi Arabia

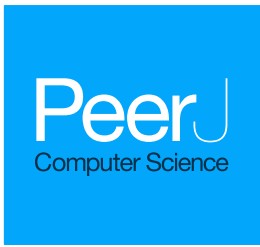

Corresponding authors
Kamran Ali Memon,
ali.kamran@kfupm.edu.sa
Khurram Karim Qureshi,
kqureshi@kfupm.edu.sa

## ABSTRACT

In the last 25 years, operators have effectively established passive optical networks (PONs), catering to around 1 billion users and earning income surpassing 8.5 billion Euros. Major standardization bodies like IEEE and ITU-T have introduced several PON solutions to mitigate last-mile broadband access and bandwidth allocation problems for end users. In this case, a compelling dynamic bandwidth allocation (DBA) algorithm can provide contention-free access (fairness) to the end user for the upstream channel with high bandwidth efficiency, minimal upstream delays, and scalability. This, in turn, boosts network quality of service (QoS) and allows operators to accommodate more users (revenue). This article examines the evolution of time-division multiplexed PON solutions such as A/BPON, EPON, GPON, XGPON, 10G-EPON, and NG-PON2 under both IEEE and ITU-T standards, addressing their approaches to DBA challenges. We analyze the bottlenecks and compare reported works based on their key strengths/applications, weaknesses, and operational mechanisms, as well as highlight their quantitative insights. We also discuss next-generation PONs (NG-PONs) and their emerging applications, such as 5G/6G fronthaul architecture in the cloud radio access network (CRAN) environment, fiber to the room (FTTR), and industrial PON, with a focus on DBA designs. Finally, the article summarizes current progress, highlights challenges, and proposes future research directions for developing more efficient DBA algorithms for these new applications.

## INTRODUCTION

Telecommunication and data networks (TDNs) have witnessed significant transformations in the last decade due to technological advancements. The demand for TDN services has risen substantially, and internet connectivity has become ubiquitous.

According to the ITU's 2023 report (*Report, 2023*), around 5.4 billion individuals use the Internet, making up 67% of the world's population, an increase of 4.7% compared to 2022. In 2023, 78% of individuals worldwide use 4G–5G mobile networks, and the average annual growth rate of fixed-broadband subscribers has been 6.7% over the last 10 years. This tremendous increase in the usage of broadband services by wired and wireless users across various applications such as social networking sites, online gaming, e-health applications, HD television, *etc.*, motivates network operators and researchers to enhance the network capabilities to meet the higher bandwidth and speed demands. Optical networks have been widely expanded into metropolitan area networks (MANs) and wide area networks (WANs) across the globe, offering high-bandwidth connectivity to telecom operators. However, there are still limitations in local area networks (LANs), and internet service providers are finding the optimal solution regarding bandwidth carrying capability and cost-effectiveness.

Traditionally, the LAN or access networks, referred to as the last mile connectivity, use the twisted pair-based copper system to provide voice and video broadband services. However, the copper-based systems have approached their upper data transfer limit, *i.e.*, the 10 Mb/s.km. To solve this problem, telecom companies put digital subscriber line (DSL) technology on top of the existing copper cable infrastructure. However, copper can carry up to 50 Mb/s of data, but only up to a certain distance—a few hundred meters between the end users and the telecom company's node (*McGarry, Maier & Reisslein, 2004*). Therefore, there is a dire need for such access technologies at LAN that can provide adequate bandwidth to high-speed Gigabit Ethernet LAN applications and satisfy the cost-sensitivity constraints of access networks (*Kramer & Pesavento, 2002*; *Memon et al., 2020*). Operators who run the public service telephone network (PSTN) decided to fix this problem by switching from a copper-based access network to a fiber-based network, also known as a passive optical network (PON). This made the fiber reach and penetration better while saving money (*Kramer, Mukherjee & Pesavento, 2002a*). Figure 1 shows the basic TDM-based PON architecture. It comprises an optical line terminal (OLT) unit, an optical network unit (ONU), and passive optical couplers, also known as splitters.

The OLT is usually placed at the central office (CO), which combines the access or end-user traffic and connects it to the core network. The ONU on the other side of the optical access link can serve one or more user interfaces. Therefore, an ONU may be placed at the consumer premises or distribution points (building or curb) along with the existing cable infrastructure (*Ma, Zhu & Cheng, 2003*). The OLT and ONUs are connected through an optical distribution network (ODN). In contrast, passive couplers/splitters combine/separate the optical signals that originated from OLT and from/to different subscriber locations, *i.e.*, toward ONUs, without needing active or expensive electronic devices such as switches, network interface cards, *etc*. In the downstream (DS) direction, OLT broadcasts a transmission signal where each ONU will receive the bandwidth grant after checking the origin address in the media access control (MAC) layer. In the upstream (US) direction, on the other hand, PON is a point-to-point (P2P) communication network that

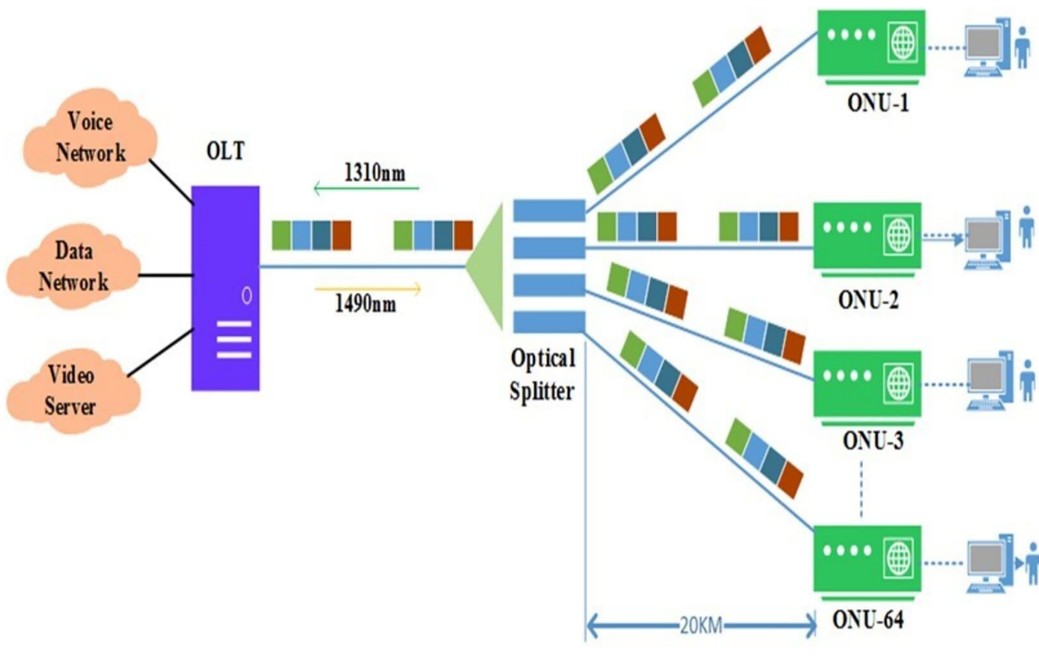

**Figure 1  Passive optical network (PON): architecture.**

mostly uses the time division multiple access (TDMA) method because OLT can not receive each ONU's bandwidth demand request at the same time on a single US wavelength.

The PON networks use two main multiplexing techniques: time division multiplexing (TDM) and wavelength division multiplexing (WDM). TDM-PONs rely on affordable hardware and existing technology. Therefore, they are the most simple and rapid methods of deploying PONs based on a technological and financial evaluation. WDM-PONs provide a consistent service with reserved bandwidth, least bandwidth assurance, and network security by giving each end user one or two operative wavelengths compared to TDM-PONs. However, they need more expensive and complicated transceivers at the OLT. Since access network technologies are cost-sensitive, implementing WDM-PON technology completely in the fiber-to-the-home (FTTH) context is difficult. Therefore, the PON operators have mostly deployed TDM or hybrid TDM/WDM technologies to scale down the cost factor in the access part of the network. Moreover, in the US direction, coordinated traffic scheduling is required between the ONUs as the distance of each ONU from the OLT changes or varies according to the geographical location of the ONU. The data bursts from these ONUs carefully require coordinated scheduling for collision-free and efficient transmission. If this mechanism is not implemented in the upstream direction, data collisions will occur, as illustrated in Fig. 2, which also causes higher propagation delays in PON transmissions. As a result, multiple access mechanisms are used in PON to avoid collisions during the bandwidth request or grant procedures (*e.g.*, US and DS), allocate the bandwidth per certain limits or rules, and reduce end-to-end delays.

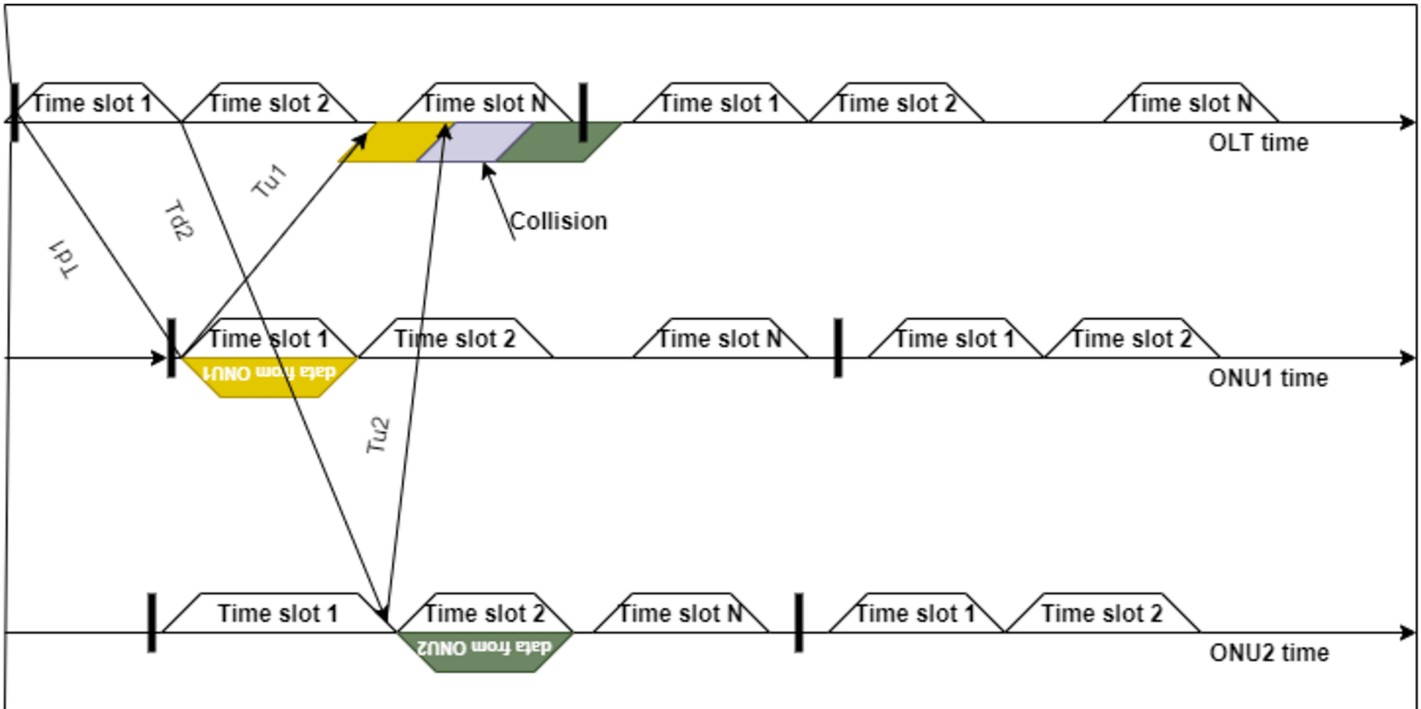

**Figure 2 An example illustration of how the data collides because of the difference in propagation delays (*Horvath, Munster & Bao, 2020*).** Td1-DS delay to ONU1, Td2-DS delay to ONU2, Tu1-US from ONU1, and Tu2- US delay from ONU2.

Various methods are studied in the literature to allocate bandwidth management in the US direction, such as the fixed bandwidth allocation (FBA) to each ONU. However, if an ONU's traffic load is minimal, this results in bandwidth waste on their part, and for an ONU with a greater traffic load, it raises queue delays. Furthermore, this method cannot allocate bandwidth to a particular traffic class inside ONU. Therefore, a dynamic bandwidth allocation (DBA) algorithm is needed to ensure each ONU gets its bandwidth and maintains the service level agreement (SLA). Using the DBA approach, service providers can earn additional revenue by serving more end users than the available bandwidth with a best-effort promise.

## PON standardization

IEEE and ITU-T are the two standardization bodies that work on developing and standardizing PONs. IEEE develops Ethernet-based PON standards like EPON, ensuring interoperability and driving innovation through research and updates. ITU-T creates global standards for various PON types based on asynchronous transfer mode (ATM)-based PON standards like APON, specifying technical requirements and ensuring global compatibility and efficiency. Both bodies have collaborated for over two decades and have developed and standardized different PON networks for application networks, *i.e.*, 1G–5G

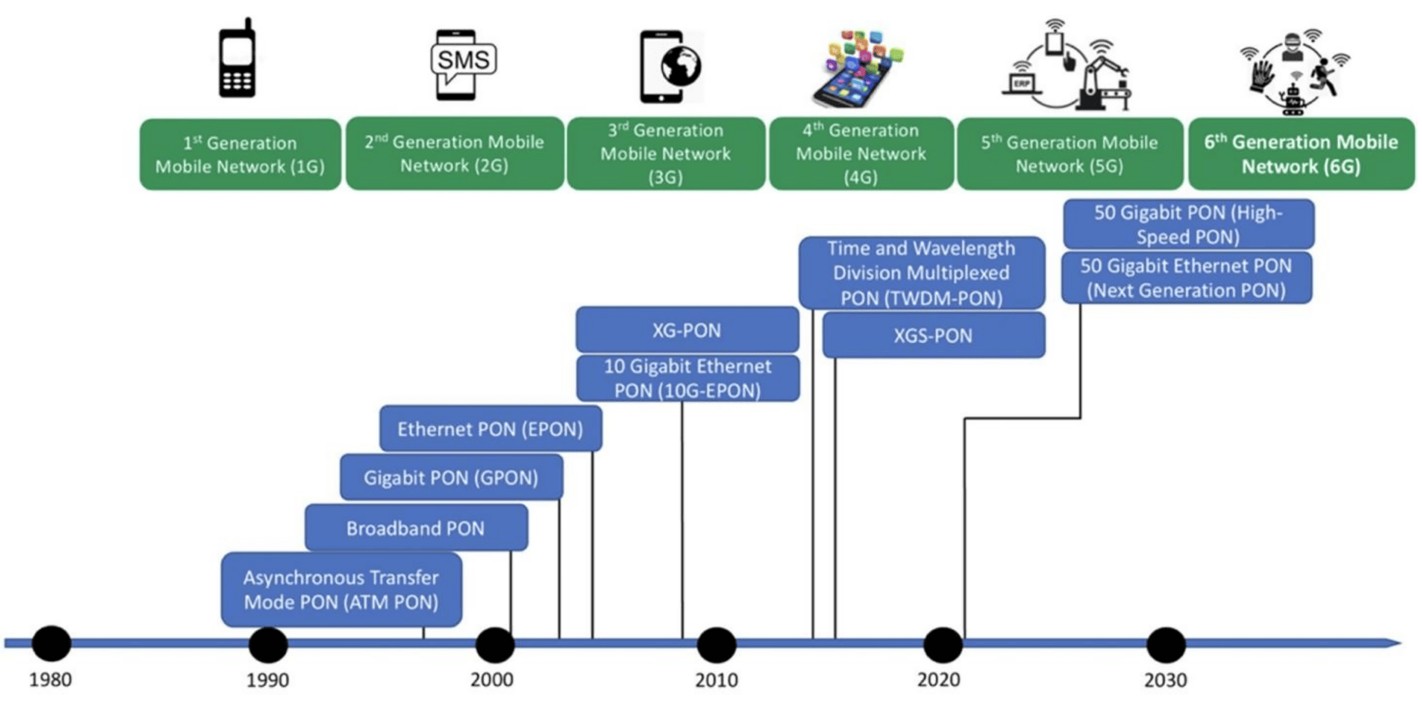

**Figure 3 Passive optical networks (PONs): evolution and application.**

**Table 1 Comparison of PON standards.**

| Parameters | APON/BPON | EPON | GPON | 10G-EPON | NG-EPON | XG-PON | NG-PON 2 |
|---|---|---|---|---|---|---|---|
| Standard | FSAN and ITU-T G.983 | IEEE 802.3ah | ITU.T G.984 | IEEE 802.3av | IEEE 802.3ca | ITU-T G.987 | ITU-T G.989 |
| Year of release | 1998 | 2004 | 2003 | 2009 | 2020 | 2010 | 2015 |
| Carried services | ATM | Ethernet and TDM | ATM, TDM, and Ethernet | Ethernet | Ethernet, TDM, WDM, CDMA | WDM | TDM, WDM, CDM |
| Data rate (Gbps) | 155 Mbps/622 Mbps | 1.244/1.244 | 2.488/1.244 | 10 Gbps | 100 Gbps | 10 Gbps | 40 Gbps |
| Split ratio | 1:16 | 1:32 | 1:32, 1:64, and 1:128 | 1:16/1:32 | 1:16–256 | 1:256 | 1:256 |
| Link speed | Average speed | Slower than GPON | Faster than EPON | Fast speed | Fast speed | Fast speed | Fast speed |
| QoS standard | ITU-T-G.983 | IEEE 802.1P | G.984 | IEEE-802.1P | IEEE-802. 1P | G 987 | −G.989 |
| Fiber distance | 20 | 10/20 Km | 20 Km | 10/20 Km | 20–60 km | 20 | 60 |
| Latency | Moderate | Low | Low | Very low | Very low | Very low | Very low |
| Delay | 2 ms | 1.5 ms | 1.5 ms | 1 ms | <1 ms | 1 ms | <1 ms |
| Cost | Lower | Lower | Moderate | Moderate | High | Higher | High |
| Implementation | Early adoption, less common now | Simple, widely adopted | Widely adopted, mature technology | Moderate complexity, suitable for high bandwidth | Complex, suitable for very high bandwidth | More complex, suitable for high bandwidth | Complex, suitable for very high bandwidth |
| Security | Basic encryption and authentication | Basic encryption and authentication | Basic encryption and authentication | Enhanced security features | Advanced security features | Enhanced security features | Advanced security features |
| Scalability | Limited | Moderate | Moderate | High | Very high | High | Very high |

(Continued)

| Parameters | APON/BPON | EPON | GPON | 10G-EPON | NG-EPON | XG-PON | NG-PON 2 |
|---|---|---|---|---|---|---|---|
| Power consumption | Moderate | Lower | Moderate | Moderate | Higher | Higher | Higher |
| Interoper-ability | Moderate, less common now | High, supported by many vendors | High, supported by many vendors | High, growing vendor support | Emerging, growing vendor support | High, supported by many vendors | Emerging, growing vendor support |
| Use cases | Early broadband access, legacy systems | Broadband access, FTTx, business services | Broadband access, FTTx, business services | High-speed broadband, FTTx, business services | Ultra high-speed broadband, advanced business services | High-speed broadband, FTTx, business services | Very high-speed broadband, data center interconnects |

mobile networks (*Union, 2013*; *IEEE, 2017*). Figure 3 shows categorization under two PON generations, *i.e.*, NG-PON1 and NG-PON2, for different application networks, *i.e.*, 1G–5G mobile networks. The ITU-T-based TDM PONs that include ATM and broadband PON (A/BPON), Gigabyte (G)-PON, XG-PON, and TWDM-PON are categorized as next-generation PON, *i.e.*, NG-PON1 and NG-PON2, respectively. The IEEE-based TDM-PONs list EPON, 10G-EPON as NG-PON1, and NG-EPON as NG-PON2. These PON standards (comparison given in Table 1) promote development, address telecommunications requirements, and guarantee compatibility, dependability, and progress in the sector.

The MAC layer structure of each PON under both standards varies and reflects the notable disparities. For example, in ITU-based PON, both OLT and ONU must transmit US/DS frames every 125 $\mu$s, *i.e.*, synchronous in nature, regardless of the traffic load. On the other hand, IEEE PONs are non-synchronous, where the OLT continuously executes the DBA algorithm for assigning bandwidth to each ONU. As a result, their DBA schemes differ and are typically incompatible. ITU PONs utilise the dynamic bandwidth report (DBRu) section of the US frame to receive queue updates from ONUs and the DS frame's bandwidth (BW) map section to allocate bandwidth to ONUs for bandwidth management purposes. IEEE PONs, on the other hand, employ a multi-point control protocol (MPCP) for this. This makes the designing of the DBA algorithm a complex task. On top of it, DBA in NG-PON 2 standards is also required to assign wavelengths, *i.e.*, dynamic bandwidth and wavelength allocation (DBWA) algorithm, monitor the ONU loads for a fair allocation of bandwidth, and reduce the wavelength switching times at OLT (*Hui et al., 2022*). This adds further to the design complexity.

## DBA design challenges for new PON applications
### PON as an optical transport solution
TDM-PON is also widely used for connecting residential and business users and serving wireless access segments like 4G and 5G, as shown in Fig. 3. It has a lot of capacity and bandwidth to be a promising optical transport solution for the fronthaul of Open Radio Access Networks (O-RAN) (*Jaffer et al., 2020*). It shows multi-tenancy, which lets different service providers or virtual network slices use the same physical infrastructure. DBA mechanisms allow resources to be flexibly distributed based on real-time demand,

benefiting the O-RAN fronthaul, where traffic patterns can vary dynamically. This technology can integrate with network slicing, creating virtual networks with specific characteristics to meet diverse O-RAN service requirements (*Wong & Ruan, 2023*). However, further research is required to develop advanced DBAs that can adapt to the dynamic nature of O-RAN architectures, optimise energy consumption, ensure security, and minimise latency.

### PON on the premises

Home internet networks must also be assured that network QoS will provide low latency and little packet loss for applications like online education and live streaming. Wi-Fi is utilised; nevertheless, it is inadequate for emerging services. Wi-Fi mesh technology enhances throughput and backhaul quality and mitigates interference. However, the restricted modulation bandwidth of copper media and air interfaces hinders high throughput and leads to inevitable collisions of packets (*Effenberger, 2022*). Fiber-to-the-room (FTTR) is an innovative fiber-based framework for home networking designed to address this issue. The goal is to ensure that gigabit connectivity is available in every corner of the home, *i.e.*, PON on the premises. In 2021, the ITU-T Q3/SG15 standardized FTTR and designated it as G.FIN (*Cases, 2021*). The development of FTTR is now in its early phase, with the introduction of novel ideas for using this technology in homes, residential regions with high population density like big apartments, and smart offices. These new PONs would support diverse applications with different bandwidth, latency, and end-to-end delay requirements.

### PON in industry

Manufacturing in industries relies on point-to-point data communications, necessitating a standardized and shared networking system. PONs are an appropriate option because they can accommodate different user interfaces and offer reliable and predictable networking, simplifying coordinated scheduling. PONs are highly effective in facilitating many manufacturing applications due to their TDMA characteristics (*Effenberger, 2024*). The European Telecommunications Standards Institute (ETSI) formed an Industry Specification Group (ISG) to precisely outline the parameters of Fifth Generation Fixed Network (F5G) to deploy industrial PON in 2021 (*Newaz et al., 2024*). The ISG provides a detailed description of the necessary specifications for PON networks in industrial environments. This includes configuring a multi-tier spine-leaf OLT architecture, the requirements for ONU interfaces, and the process for allocating bandwidths. AI is regarded as a fundamental component in the F5G architecture for network automation (*ETSI Recommendations, 2023*). Active research is being conducted on using PON during manufacturing to address the need for low latency and jitter at the millisecond level or below to fulfil industrial needs.

### Contributions and layout of the article

Even though the recently proposed DBA/DBWA algorithms are deemed adequate, more new methods will continue to be proposed. We cannot presume that an ideal bandwidth allocation technique can consider every parameter. Some could be less complicated, some

might have lower access delays, and others might have better bandwidth consumption. The MAC engine or the frame-level structures used for US and DS communication in any DBA/DBWA reflect the features and complexity of the PON that the algorithm offers. As a result, it is of prime concern to evaluate and understand the differences in features, limitations, and MAC engine designs along with architectural needs/progress in the present and latest PON variants/applications in use for 2.5 decades. We present the following contributions:

- First: This article explains the primary PON variants, their development up till now, architecture, and operating procedures. In continuation, particular emphasis is given to the working of the DBA MAC engines for various TDM-PONs that follow ITU-T and IEEE standards and end the section with the present architectural challenges.
- This article presents a critical review of the well-known DBA algorithms for the TDM PONs, reported in the literature in terms of the key strengths/weaknesses and their quantitative insights.
- We present the analysis on implementing TDM PON as an optical transport solution for 5G mobile fronthaul (MFH) network and present DBA requirements and challenges.
- We also look at the newest PON applications and give DBA algorithm designers goals and steps they can take to deal with future problems related to quality of experience (QoE), allocation of resources, and quality of service (QoS) for time-sensitive applications, supporting industry 4.0 and IoT-driven automation in these areas.

The rest of the article is organized as follows: Methodology presents the methodology, Architectural and Technological Advancements in PONs discusses the PON variants with the latest architectural developments and challenges, and how standard DBA algorithms have changed over the past 25 years under ITU-T and IEEE standards. Dynamic Bandwidth Allocation Algorithms of TDS-PONs thoroughly evaluates the reported DBA schemes for EPON, GPON, 10G-EPON, and NG-PON 2. Next, Evolving PON Applications and Key DBA Challenges presents the potential applications of PONs *i.e.*, TDM PON-based 5G fronthaul network, FTTR, and Industrial PONs, and reports the challenges, opportunities, actionable steps, and targets concerning the DBA algorithms for each, while the Conclusion concludes the article.

## METHODOLOGY

This survey follows a systematic literature review approach focusing on DBAs in the different PON variants across the 2.5 decades. This article highlights the latest architectural developments with its issues and new PON applications with recent research trends.

### Explored topics

ET1: What are the PON DBA structures?

Answer: Standardization bodies have proposed the different PON variants which include APON/BPON, GPON, EPON, XG-PON, 10G-EPON, and NG-PON 2 *i.e.*, TDM PONs. The DBA frame structure for each PON variant varies in operation and features.

Many researchers aim to understand both the architectural development and DBA structures through standard reports.

ET2: What are the key performance indicators or comparison factors to evaluate these different TDM PON DBAs proposed so far?

Answer: The key performance indicators proposed for DBAs include algorithm complexity, throughput, and end-to-end delay. "Dynamic Bandwidth Allocation Algorithms of Tdm-pons" presents the summary and the comparison of the reported DBAs for each PON variant.

ET3: What is the role of the PON DBAs in the 5G MFH networks and related challenges? Answer: PON DBA plays a pivotal role in optimizing MFH networks, ensuring low latency, efficient bandwidth utilization, and seamless connectivity for 4G/5G applications and future 6G. "Evolving Pon Applications and Key Dba Challenges" addresses this in detail.

ET4: What are the open research issues for the new PON applications such as FTTR and industrial PONs?

Answer: Open issues include low latency and minimal roaming delay, deployment costs and standardization for FTTR and synchronization, time-sensitive networking, QoS, and power efficiency for industrial PONs. The detailed discussion is presented in "Conclusion".

ET5: Are there any active research projects focusing on the new PON architectures and associated open research issues? Does the PON research community still have a growing research interest in DBAs?

Answer: Yes, subsection 6 explores this topic. Engineers, researchers, vendors, and policymakers are increasingly interested in understanding DBA algorithms and their influence on network performance. This article offers significant insights that enrich their comprehension of DBA algorithms and their practical ramifications in real-world networks.

## Paper selection criteria, keywords and databases

This article has targeted well-known journals and conferences published in the English language from 1997 to 2024, addressing the different DBA schemes for PON variants and applications. Published articles were searched by these keywords "Dynamic bandwidth allocation" or "DBA" or "Passive optical network" or "Optical Access Network" or "PON" or "OAN". The reported well-known DBAs were chosen only from Google Scholar, Elsevier, and IEEE libraries.

## ARCHITECTURAL AND TECHNOLOGICAL ADVANCEMENTS IN PONS

Figure 4 shows the classification of PONs in TDM, WDM, and combo PONs. Table 1 presents the comparison of all the PON standards. The combo PONs combine the different PON technologies to achieve flexibility and scalability in network deployment and optimize network resources. The architectural and technological developments are

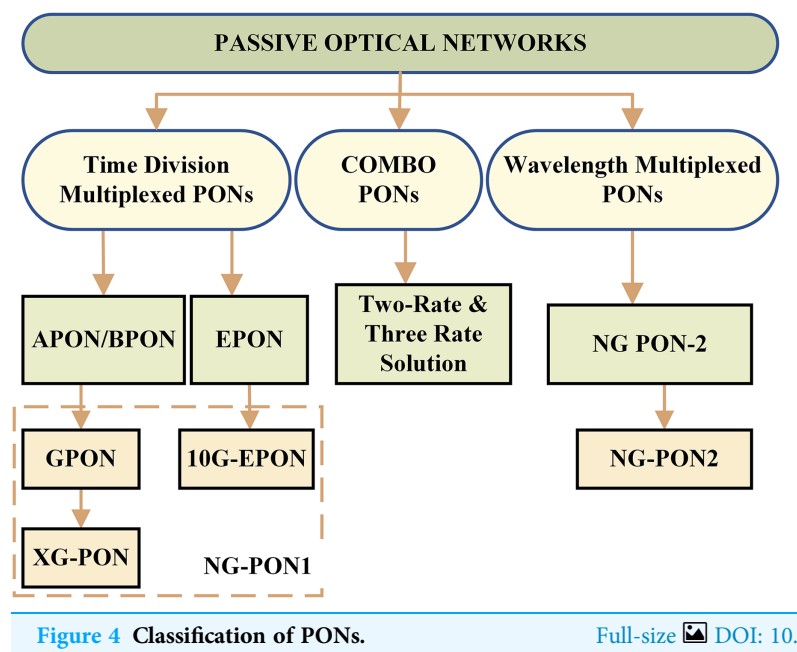

**Figure 4 Classification of PONs.**

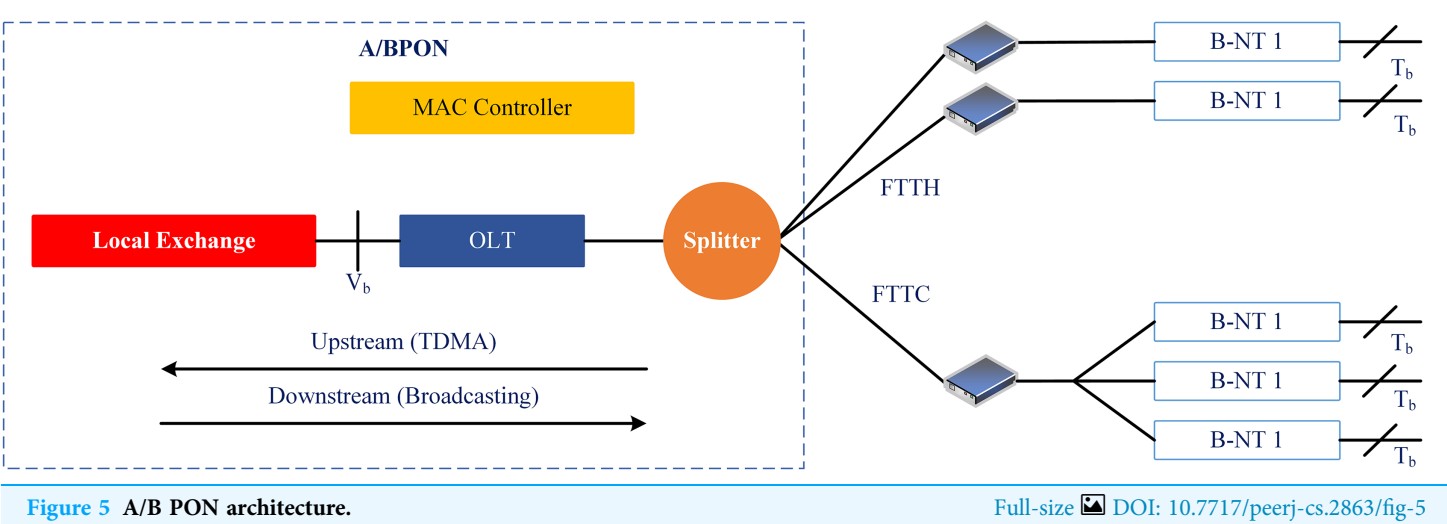

**Figure 5 A/B PON architecture.**

highlighted, and the frame structures and US-DS DBA mechanisms of TDM PONs for the two generations are discussed below:

## NG-PON 1 standards

### APON/BPON-ITU-T Rec. G.983.x (1998/2001)

ATM PON (APON) is the first PON architecture to guarantee quality of service for both voice and data services at the same time (*Kani & van Veen, 2020*), as shown in Fig. 5.

The ATM frame, shown in Fig. 6, comprises a concatenation of 56 ATM cells of 53 bytes. The 1$^{st}$ and 29$^{th}$ ATM cells are Physical Layer Operation Administration and Management (PLOAM) cells used for cell separation and synchronization in the receiver

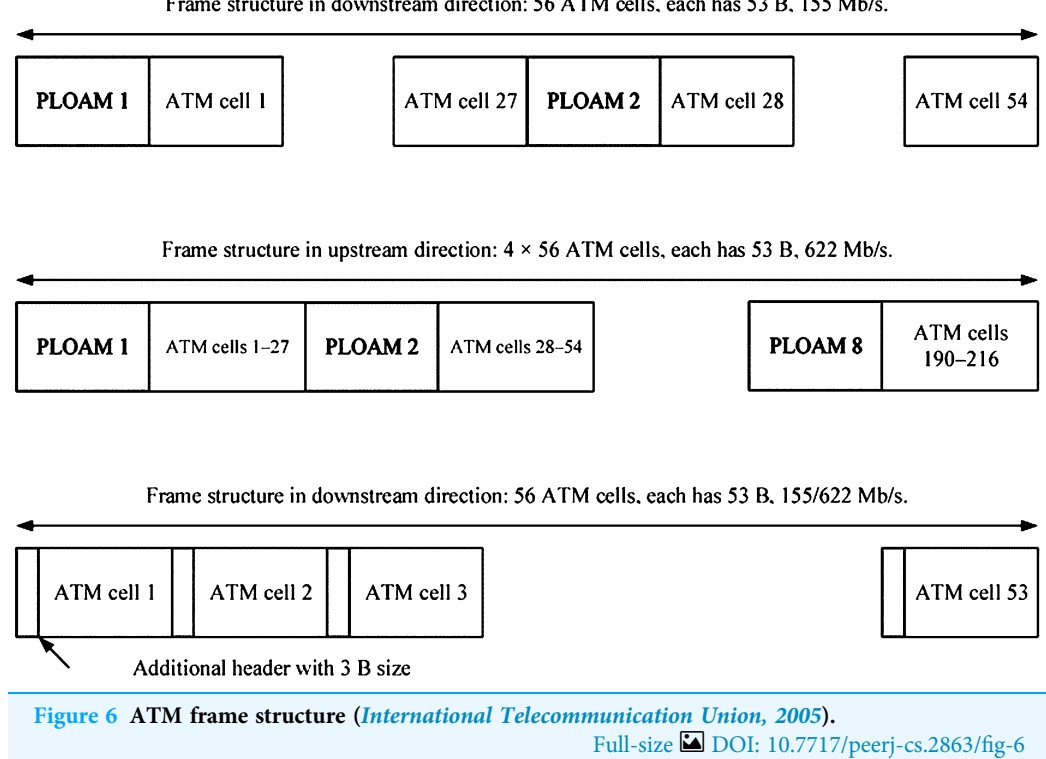

**Figure 6** ATM frame structure (*International Telecommunication Union, 2005*).

and support transfer rates up to 1.5 Mb/s to 2.4 Gb/s. In the US, OLT receives the queue size information *via* PLOAM messages from ONUs and allocates the required bandwidth accordingly. In DS, ONUs receive the grant *via* PLOAM messages from OLT. The grants act as permissions for the ONUs to send user information to ATM cells; this can impact how bandwidth is allotted to ONU units. APON supported the transmission rate of 4.8 Mb/s in both US/DS for 32 end users, following FBA with a transfer rate of 155 Mb/s.

BPON, an improved version of APON, introduces the DBA concept under G.983.4 recommendations to benefit from the remaining or unused bandwidth of the users. The initial DBA mechanism uses three distinct techniques (*Uzunidis et al., 2022*):

I. A status report (SR) only;
II. The status report itself;
III. A mixed report.

In the status report only, the OLT observes the traffic continuously and assesses this occurrence as a need to raise the bandwidth if the occupancy of queues rises. The ONU notifies the OLT of its status in the second scenario. The OLT receives a request from the ONU for greater bandwidth (when required). A mixed report combines elements of the two earlier categories. Table 1 compares the PON standards regarding the service, data rates, grant schedule, split ratio, link speed, and many other important factors for critically evaluating architectural and technological developments.

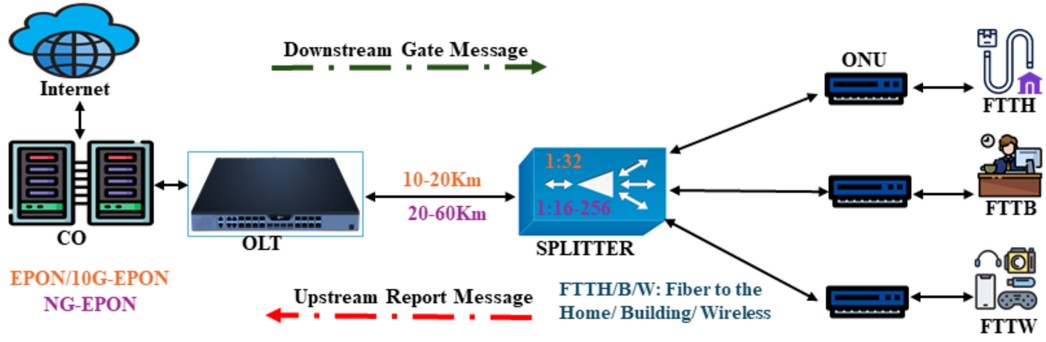

**Figure 7** DS and US transmissions in EPON/10G-EPON.

### EPON-IEEE 802.3ah (2001)

In 2004, IEEE introduced Ethernet-PON (EPON), using Ethernet as a transport protocol and providing symmetric 1.25 Gb/s with variable-length Ethernet frames (packets) up to 1,518 bytes. Ethernet technology is a simple and low-cost solution (cheap fabrication costs) and has almost covered the network market. In EPON, OLT is connected to the multiple ONUs *via* a 1:N optical coupler or splitter, as shown in Fig. 7. So, each ONU buffers data obtained from the OLT in the DS. The round-trip time (RTT) between the OLT and each ONU also varies due to the differing link lengths between the OLT and each ONU. In the US, ONUs are unable to communicate with each other directly. Instead, each ONU can only transfer data to the OLT.

Consequently, in the DS, an EPON may be observed as a point-to-multipoint network and in the US as a multipoint-to-point network (*Li, Lv & Bi, 2024*). Different methods can be applied to avoid data collisions. For instance, wavelength division multiplexing (WDM) PON can be used. However, this approach is deemed costly because the OLT necessitates a dedicated receiver or an array of receivers to receive data across various wavelength channels, and each ONU must also be outfitted with a transceiver tailored to the wavelength. Table 1 presents the technical comparison of EPON with other standards.

Figures 8A and 8B show that the Ethernet frame is converted into an EPON frame by replacing its preamble with a logical link identifier (LLID) ahead of the frame header that comprises the destination address (DA) and source address (SA). Therefore, the MAC mechanism is required to prevent collisions and allocate bandwidth fairly among the shared connection capacity. Users must be in sync and aware of the permitted transmission time window. Each time slot may carry a constant or variable number of Ethernet frames. When the designated time slot for a certain user occurs, data (in the form of Ethernet frames) is delivered in bursts. The user transmits an empty frame if there are no contents. Customers can benefit from upgraded services that require flexible data rates that cannot be legitimately given under FBA, *i.e.*, the PON can serve additional users using a DBA mechanism. It determines how long each user creates a queue to keep data. The OLT is responsible for gathering details on queue length and allocating timeslots to individual users. The approach promotes the effective use of the shared link capacity

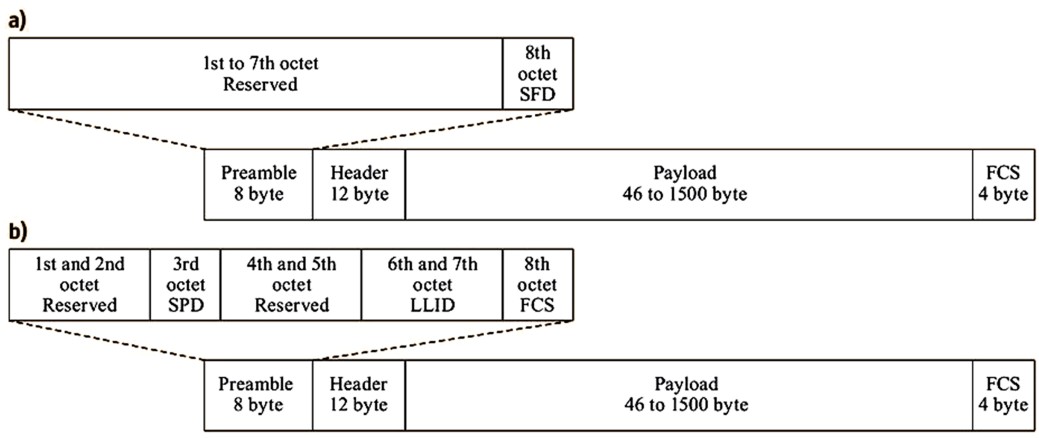

**Figure 8 Ethernet and EPON Frame (*Chae, Wong & Tucker, 2002*).**

despite the higher transmission overhead. The multi-point control protocol (MPCP) is a signalling protocol at the MAC layer. It facilitates communication between the OLT and ONUs/ONTs for inter-ONU scheduling and service class support. The three functionalities of MPCP enable PON vendors to apply the preferred DBA system easily:

**Discovery processing**: PON users are registered and detected through a three-way handshaking process. In the process, the ONU generates the REGISTER REQ message, the OLT responds with the REGISTER message, and the ONU then echoes the registration parameters back to the OLT *via* the REGISTER ACK message.

**MPCP report handling**: ONUs report their status or bandwidth requests using the MPCP REPORT message, which contains information regarding the amount of data waiting to transmit a specific LLID.

**MPCP gate handling**: OLT polls ONUs for their queue status and grants bandwidth using the MPCP GATE message.

Interleaved polling with adaptive cycle time (IPACT) was the first EPON DBA mechanism to evaluate the time slot duration of each ONU. Several variations were also proposed (*Logothetis et al., 2013*), but overall, till this end, DBAs were not using the priority queues to offer differentiated services.

### Gigabit PON (GPON): ITU-T Rec. G.984.x (2004)

The ever-increasing demands for bandwidth by users trigger the development of new PON standards. The GPON is an extended version of the BPON and uses the GPON encapsulation method (GEM) to support both variable-length and fixed-length forms of service, for instance, Ethernet and ATM. By using GEM frame fragmentation at the point of sending, GPON effectively lowers latency and facilitates real-time transmission. The GPON offers service providers the ability to retain conventional services (TDM-based voice, leased lines) without the need to replace customer peripheral equipment (CPE). This standard delivers 1.2 and 2.4 Gbps US/DS for a network distance of 20 km. Table 1 presents the technical comparison with other standards.

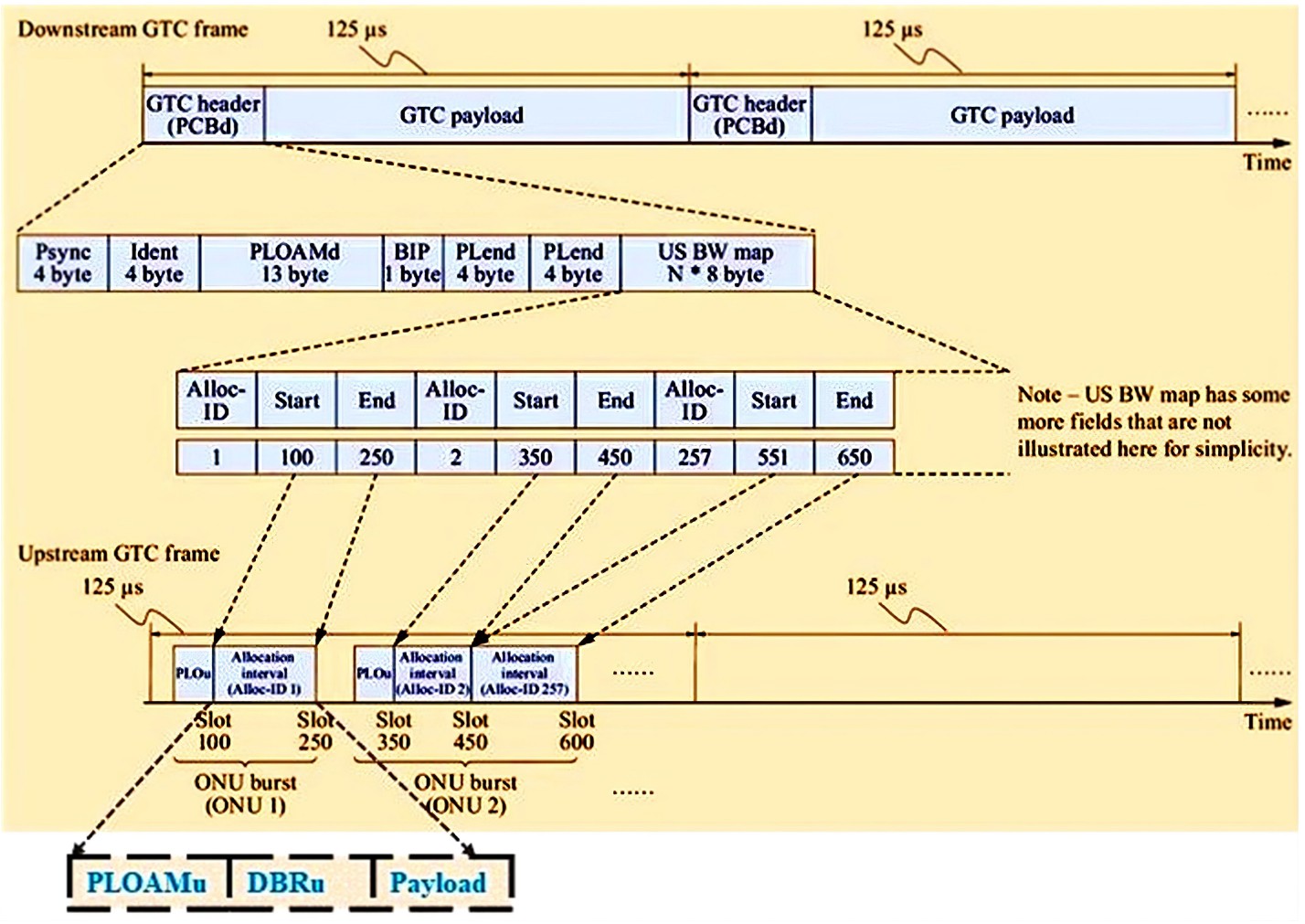

**Figure 9  GTC frame structure for US and DS with ONU burst.**           

GPON employs TDM and TDMA mechanisms in DS and the US, respectively. These are executed in the GPON Transmission Convergence (GTC) layer for frames with a constant frame duration of 125 ms (*Thangappan et al., 2020*). To guarantee that the transmitting frame aligns with the designated time slot in the TDMA process, synchronization is necessary in the US due to the unequal distances between ONUs and OLT. To achieve this objective, the OLT establishes communication with each ONU and, taking into account the RTT, calculates a delay for each ONU in reverse order of the RTT. This ensures that the propagation delay between the OLT and ONUs is balanced. The DS_GTC frame, as shown in Fig. 9, consists of a physical control block downstream (PCBd) for the synchronisation and payload. Other fields in PCBd include:

- US bandwidth (BW) map-for start and end time information for each allocation interval.
- Physical synchronisation (Psync)-for frame synchronisation
- IDENT-for forward error correction

**Table 2 T-CONT types.**

| T-CONT # | Definition | Application |
|---|---|---|
| 1 | Fixed bandwidth (offered without demand) for services sensitive to delay | VoD, VoIP |
| 2 | Assured bandwidth (offered after demand) for services insensitive to delay | Data transfer |
| 3 | Combination of Assured and non-assured bandwidth (if available but not guaranteed) | Triple play services |
| 4 | Best effort bandwidth (if remaining uplink bandwidth is available; not guaranteed) | Best effort |
| 5 | Combination of all T-CONTs | – |

- Physical-Layer Operations, Administration, and Maintenance for DS (PLOAMd)-contains operation, administration, and maintenance commands.
- Bit-interleaved parity (BIP)-carries FEC parity for detecting error(s) in reception.
- Payload length downstream (PLend)-specifies the length of the preceding US BW map field.

On the other hand, the US_GTC frame consists of a burst of time slots called the transmission container (T-CONT). These are distinguished in the GPON using a specific identifier known as Alloc-IDs. A T-CONT solely comprises either ATM cells or GEM frames, but not both. Each ONU allocates traffic flows with identical attributes to one or more T-CONT types out of five possible T-CONT types, given in Table 2.

The ONU burst structure in Fig. 9 illustrates that the ONU relies on the Dynamic Bandwidth Report US (DBRu) to tell the OLT how much traffic is waiting to be sent in a certain Alloc-ID. Then, OLT checks the reported bandwidths against each Alloc ID to give grants automatically in the next allocation intervals based on the SLA and the DBA that was used (*Abbas & Gregory, 2016*).

### 10G-EPON-IEEE 803.2av (2009)

10G-EPON can support the vast number of consumers in metropolitan areas with broadband connectivity, maintaining backward compatibility with EPON. To make 1G E-PON and 10G E-PON work together again through TDM, the US wavelength region for 10G E-PON can be lined up with that of 1G E-PON. 10G-EPON offers 1 and 10 Gbps US and DS data speeds over 20 km. In addition to high reach, the low cost per gigabit per second of bandwidth and providing bandwidth-intensive services such as Triple Play Service, *i.e.*, audio, video, and data, are other pluses. Figure 7 shows the architecture of the US/DS transmissions in 10G-EPON. Table 1 lists the most improved features over time.

In the same way that EPON does, 10G-EPON uses MPCP and the DBA algorithm to support QoS. At the link layer, 802.1Q divides network traffic into eight groups. Each category has a separate queue in each ONU with six priority levels for user traffic (*Uzunidis et al., 2022*). The system handles time and safety-critical traffic, inter-network control, voice, video, controlled load, excellent effort, best effort, and background traffic. The IEEE 802.3av task force focused on the physical layer, leaving the MAC protocol unchanged. The 10G-EPON MAC-layer control protocol includes enhancements for managing 10G-

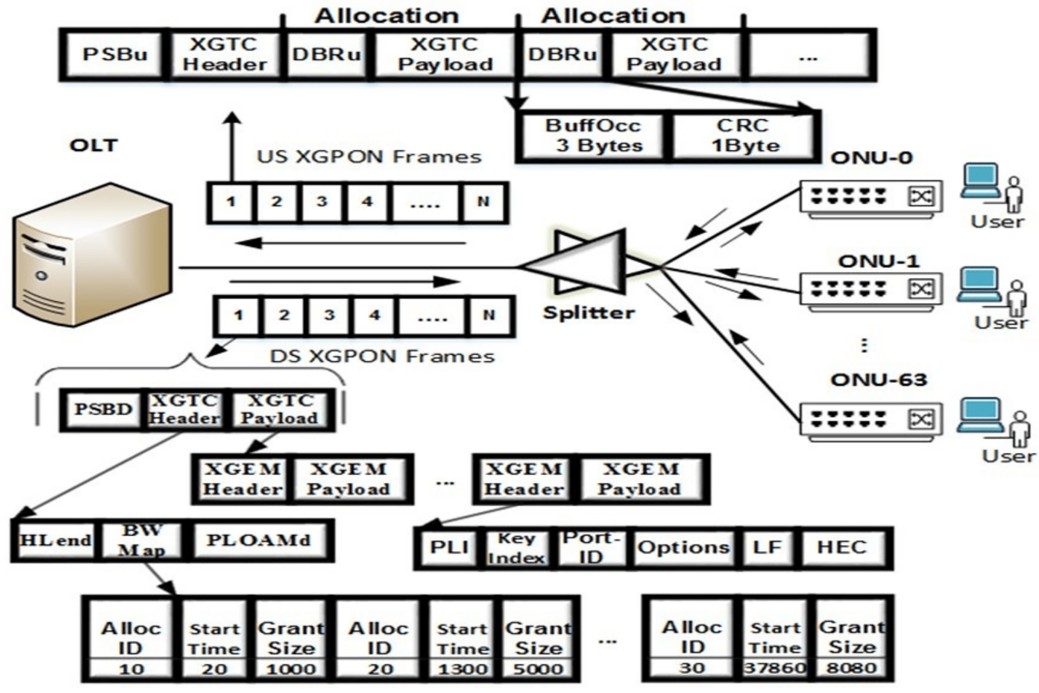

**Figure 10 XG-PON: architecture and DBA process (*Butt et al., 2018b*).**

EPON FEC with Reed-Solomon coding and inter-burst overhead, but it results in up to 12.9% overhead (*Laboratory, 2024*).

### XG-PON-ITU-T Rec. G.987.x (2010)

A total of 10 Gb/s GPON (XG-PON) was developed to accommodate emerging bandwidth-demanding applications, connect more users, and improve QoS and security. XG-PON delivers asymmetric transmission at 10 Gb/s in DS and approximately 2.5 Gb/s in the US. Like GPON, XG(S)PON supports ATM and Ethernet traffic *via* the XG-PON encapsulation method (XGEM). ONU registration and ranging mechanisms are handled through the DS and US PLOAM sections, and OLT can send several PLOAM messages (specified in the Hlend segment by the PLOAM count field) in a DS frame. Likewise, the bandwidth assignment (Grant-Request) method is incorporated in the DS frame of the OLT scheduler *via* the bandwidth map (BWmap) segment and the DBRu segment in the US frame. Figure 10 shows the architecture and DBA cycle of the XG-PON, whereas Table 1 compares the specifications with other standards.

### NG-PON 2 standards

### NG-EPON-IEEE 803.2ca (2020)

As communication requirements and technology advance, the need for a new PON standard with better specifications arises. An increasing number of users require additional bandwidth for various tasks, such as wireless fronthaul. At the same time, there is a need to increase the number of users per fiber for commercial purposes. Another motivating factor

is accomplishing these goals at a more affordable price than NG-PON2. A new EPON standard (NG-EPON) emerged in 2020 (*Lam & Yin, 2020*). This system would function at combined data rates of 100 Gb/s to offer increased data speeds for each user. Four wavelengths with a combined capacity of 25 GB/s can also be multiplexed to achieve this. The IEEE task group is now focused on developing three distinct generations of the access network, *i.e.*, 25, 50, and 100 Gb/s. The architecture displaying the US/DS transmissions in NG-EPON is shown in Fig. 7, and the comparative advancements of the features are given in Table 1.

The Data Link Layer's multi-point MAC control (MPMC) sublayer is responsible for adding the NG-EPON to the Ethernet architecture and ensuring it stays compatible with older IEEE standards. It includes protocols for transmission resource allocation, ONU discovery and registration, and DBA. Dynamic wavelength channel bonding is used to achieve high data rates. The Multi-Channel Reconciliation Sublayer (MCRS) facilitates data transmission over numerous signals across multiple MCRS channels. The fundamental transmission unit is the envelope quantum (EQ), which comprises 72 bits. Frames can be categorized as either complete or fragmented. The transmission of a signal involves the interleaving of frames across multiple MCRS channels to obtain higher transmission speeds. The transceiver class utilizes 25 Gb/s avalanche photodiodes (APD) for receivers, employing NRZ modulation. The optical modulation amplitude (OMA) minus penalty approach is used for transmitters. Currently, Optical Network Units (ONUs) require burst-mode laser diodes equipped with a rapid on-off switch and a control loop to regulate the laser bias current (*Uzunidis et al., 2022*). The NG-PON2 transmission convergence (TC) layer allows for operations with more than one wavelength and wavelength channel mobility, which means it can switch from one TWDM source channel to another target channel.

### NG-PON2-ITU-T Rec. G.989 (2015)

NGPON2 is a hybrid approach based on time and wavelength division multiplexing (TWDM) techniques. The ITU-T and FSAN jointly developed this hybrid PON standard, providing over 40 Gbps to meet such relentless growth in traffic demand with low-cost TDM equipment. The NG-PON2 can easily co-exist with the existing ODN infrastructure of the previous generation of PON, such as the EPON, GPON, and XG(S)-PON, attracting the telecom operators for quick deployment in 2017. Moreover, the PON industry has also suggested NGPON2 technology by using the TWDM approach, not only to support fixed broadband user data but also to provide 5G wireless service. Therefore, NGPON2 is considered the ideal technology for connecting the fronthaul/backhaul link of the small cell architecture of 5G.

Figure 11 presents the architecture of TWDM-PON, wherein an OLT has optical subscriber units (OSUs), a wavelength multiplexer (to keep multiple wavelengths in a single ODN featuring tunable transceivers), a passive splitter, and several ONUs. Table 1 presents the comparison with other PON standards in terms of the specifications. The tunable transceivers offer high flexibility and reconfigurability, allowing them to communicate with any OLT transceiver at any wavelength. This colorless design simplifies

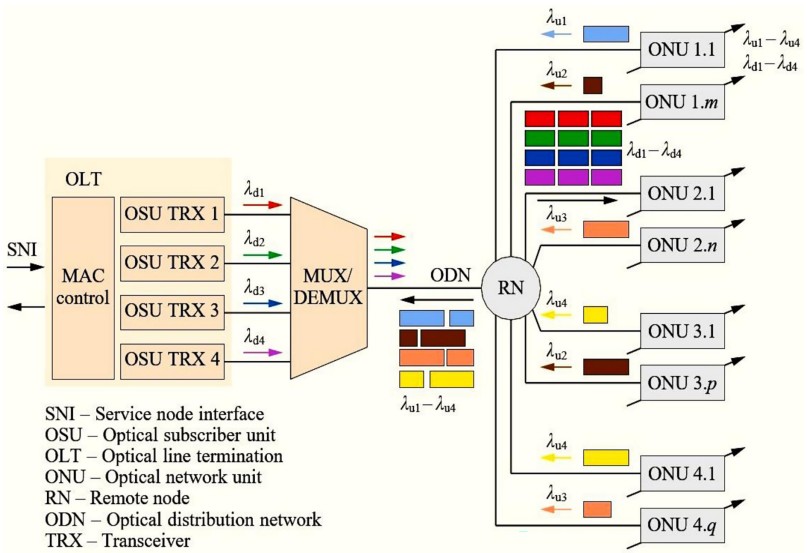

**Figure 11** NG-PON2 architecture (*Prat & Valcarenghi, 2020*).

ONU installation and management (*Memon et al., 2020*). Under the MAC layer, this standard provides two channels, *i.e.*, TWDA channels (TDM/TDMA access—P2MP) and WDM channels (P2P access), *i.e.*, development of the dynamic bandwidth and wavelength algorithms (DBWA) (*ITU-T Recommendations, 2021*).

This is possible with OLT channel terminations (OLT CTs) communicating *via* the inter-channel termination protocol (ICTP). Referring to Figs. S1 and S2 (*ITU-T Recommendations, 2021*), it can be seen that the TWDM transmission convergence (TC) layer is made up of three sublayers: the service adaptation (SAS) layer, the framing (FS) layer, and the PHY adaptation (PS) layer.

Every ONU begins a US PHY frame by adjusting the starting position of the DS frame. The OLT CT and all ONUs on the PON share a common timing reference. In the US, each ONU transmits short PHY bursts with PSBu and payload (as Framing Sublayer), remaining silent between them. The OLT CT uses guard time (TX enable/disable times, 64-bit) and BWmap to control the timing and duration of these bursts to avoid overlap, collision, or jamming in US transmissions of different ONUs. The SAS uses XGEM for service data unit (SDU) encapsulation, representing its protocol data unit (PDU) as an XGEM frame. The XGEM header consists of an eight-byte encrypted field that specifies the logical connection or traffic pattern, whereas the XGEM payload might include a full TC layer SDU. In TWDM-PON, US bandwidth is assigned using a BWmap, transmitted every 125 μs and containing allocation structures (AS) for a particular ONU. Alloc-ID serves as the grant recipient's identification, and T-CONTs are in charge of managing it. A 1-bit forced wakeup indication (FWI) flag is used for energy-efficient scheme operations. The MAC layer uses an extra auxiliary management and control channel (AMCC) for dynamic wavelength assignment, reusing the TWDM TC's three layers. FEC is subject to ON/OFF control for TWDM channels and is integral to PtP WDM channels. NG-PON2 uses the

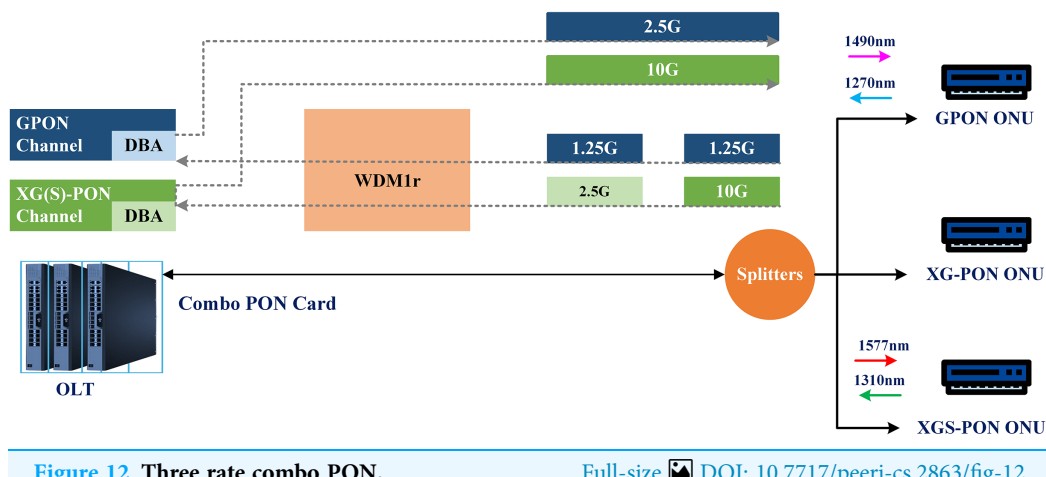

**Figure 12  Three rate combo PON.**               

PLOAM messaging channel like XG-PON, with additional features like wavelength handover, protection, and power control.

## Combo PONs

The growing popularity of bandwidth-intensive video services such as 4K/8K and AR/VR drives the need for more network capacity. As a result, there is a significant increase in the building of large-scale PON networks in the optical access network. Nevertheless, to incorporate various NG-PON1 standards, such as GPON-XGPON and EPON-10G-EPON, the smooth evolution of the network must consider the need to utilize the existing ODN and ensure forward compatibility with legacy ONUs. This places higher demands on the PON technologies (*Ding et al., 2022*). ZTE introduced the industry's first Combo PON solution, combining XG-PON and GPON for 10G-GPON construction. This two-rate solution is popular due to compatibility, ease of use, and initial expenditure protection. ZTE also introduced a three-rate solution, combining XGS-PON, XG-PON, and GPON (see Fig. 12), facilitating smooth evolution from GPON to XGS-PON (*Guanqiao, 2021*).

GPON and XGS-PON signals are transmitted and received separately in an optical module using a three-rate Combo PON. The module's embedded multiplexer, WDM1r, combines and divides the four wavelengths needed by both GPON and XGS-PON signals. The three-rate Combo PON is a solution that offers several advantages over the PON solution using an external optical multiplexer.

- It does not require adjustments to the ODN, making deployment easier (*Kim, Doo & Chung, 2023*).
- The solution does not introduce new insertion losses, which can strain the existing optical power budget.
- The same-class optical module of the Combo PON does not change the optical power budget margin of the ODN.

- Additionally, equipment room space is saved, and network O&M is simplified, as the optical module integrates XGS-PON/XG-PON/GPON and WDM1r functions, eliminating the need for additional equipment and reducing O&M complexity.
- The service provisioning process remains unchanged, and service migration is fast due to the easy interconnection between the element management system (EMS) and operation support system (OSS), which utilizes existing interconnection modes.

Photonic integrated circuit technology (PICT) provides a power-efficient solution for next-generation PON transceivers with many ports. Despite being cost-sensitive, the market size of PON devices is projected to see significant growth, leading to a decrease in device costs and an acceleration in the development of PICT. Currently, research is being carried out to integrate two and three generations of PONs (*Jin et al., 2023b*).

### Recent advancements
ITU-T released the most recent PON specifications under the Higher Speed PON (HS-PON) project, which defines a 50G-PON system with a DS of 50 G and a US of 25 G. In February 2023, two revisions to the HS-PON standards were officially authorized, signifying that the system specifications have reached a level of maturity suitable for commercial development and implementation by the telecom operators at the beginning of 2025 (*ITU-T Recommendations, 2023*). HS-PON employs electronic equalization to counteract chromatic dispersion (CD) and utilizes the O-band, the region with the least dispersion. Nevertheless, the challenge is observing forthcoming transmitter advancements due to the high signaling speeds above 50 Gb/s.

### Latest architectural challenges
ITU-T began a project on very high-speed PON (VHSP) systems to achieve above 50 Gb/s per wavelength. This project intends to collect the VHSP system requirements, characteristics, and prospective technologies and pave the way for the standardization of future PON systems after the successful implementation of 50G-PON (*ITU-T, 2024b*). IM-DD, which utilizes power budget improvement technologies to increase performance, and coherent technologies (CTs), which use coherence detection to address fiber dispersion, are two candidate technologies that are now the subject of considerable debate over their potential applications.

- The most significant obstacle that IM-DD must overcome to achieve high-speed transmission is fiber dispersion, which results in the deterioration of optical signals and a decrease in receiver sensitivity. At 100 Gb/s, the receiver's sensitivity would drop from −27 dBm at 50 Gb/s (*Rosales et al., 2022*) to −20 dBm (*Yi et al., 2019*).
- CTs can use DSPs to mitigate these degradations but at the cost of increased system complexity, higher costs, and power consumption (*Rizzelli, Torres-Ferrera & Gaudino, 2023*). Companies such as British Telecom, Huawei, ZTE, and others are now participating as members and providing support.

# DYNAMIC BANDWIDTH ALLOCATION ALGORITHMS OF TDM-PONS

In PON, the OLT is linked to the backbone of the PSTN network and the IP network. Through this interface, the OLT is designed to serve native PSTN and IP-based services, such as browsing and streaming YouTube or Netflix channels to download pictures or videos from these sites. The native PSTN service requires static or constant upstream traffic, whereas, for IP-based service, the traffic in the upstream channel is dynamic or bursty by nature. In the past, network usage was asymmetric–downstream required much more bandwidth than upstream, and now, this trend is changing. As seen in Fig. 3, upstream bandwidth consumption increases over time. Therefore, the service provider needs efficient bandwidth allocation schemes that dynamically allocate bandwidth to the end nodes according to their requirements. The DBA dynamically allocates bandwidth to ONUs according to the required traffic in the upstream direction. The OLT assigns bandwidth to ONUs in PON, so the DBA is under OLT control.

## DBA algorithm for EPON

The DBA module uses the DBA algorithm to compute the collision-free US transmission schedule of ONUs and produce GATE messages accordingly. We have categorized the DBA algorithms for EPONs into algorithms without differentiated QoS support and algorithms with differentiated QoS support. Here, we will discuss the DBA algorithms of each method in detail:

### *DBA without differentiated QoS support*

*Luo & Ansari (2005)* in proposed the DBA algorithm based on the volume of each ONU buffer or request. After the ONUs request, the OLT plans the time slot in which ONUs can communicate several bytes. The authors considered the tree architecture in the proposed setup, where the three ONUs are connected to the single power splitter. The interleaved polling with adaptive cycle time (IPACT) complete operation steps are shown in Fig. 13 (*Zheng & Mouftah, 2009*).

The algorithm's first step is to know the waiting number of bytes in every ONU and round trip time (RTT) and the total number of bytes in ONUs, where a polling table is stored at OLT. The OLT also sends a GRANT message to ONU1 at some time T1 to start the 3,200-byte transmission. In the next step, after receiving the GRANT message from OLT, ONU1 will start uploading its data; hence, the ONU has to continue buffering and receiving packets from end subscribers, so ONU1 will produce a REQUEST noting how many bytes are kept in its buffer. After that, OLT keeps tracking the REQUEST data, and after its arrival, it updates the polling table and calculates the succeeding time slot. After that, 3200 bytes were changed to 1,200 at time T1.

Now, at time T2, the ONU2 will start data transmission, sending and receiving the frame from the OLT after receiving the GRANT note, so from the above explanation, it is understood how IPACT repeats the same process to ONU3 at time T3. According to the explanations and the steps above, we can conclude that the DBA algorithm will be defined as following (*Zheng & Mouftah, 2009*).

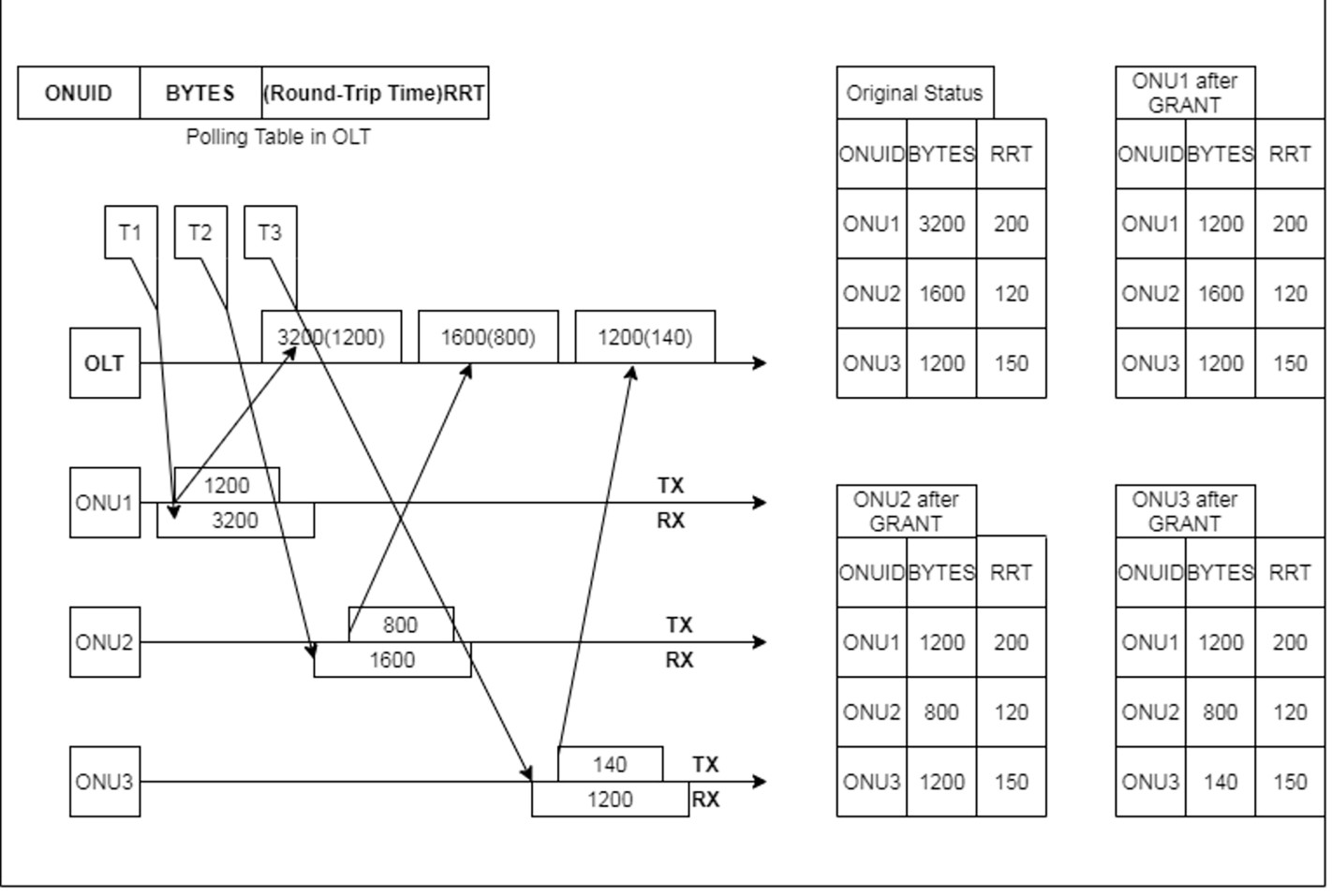

**Figure 13** IPACT algorithm: operation steps (*McGarry, Maier & Reisslein, 2004*).

$$T_k^{i+1} = \max \begin{cases} RTT_i + T_k^i + T_{guard} + \frac{W_{grant}}{R}, \\ -RTT_{i+1} T_{i+1}^{k-1} + RTT_{i+1} \end{cases} \tag{1}$$

where $T_k^i$ represents the start time of $ONU_{i+1}$ transmission in the $k$-th cycle, $W_{grant}$ represents the grant window for each ONU, $R$ is the transmission bit rate (bits/sec), $T_{guard}$ is the guard time between transmissions and $RTT_{i+1} T_{i+1}^{k-1}$ represents a correction factor.

### Bandwidth guaranteed polling algorithm

There are many deficiencies in IPACT. It cannot make differences between numerous classes of services, and it does not support priority scheduling schemes because it follows first-in-first-out transmission. A new algorithm has been proposed to overcome these issues, which joins the novel IPACT restricted-service system named inter-ONU scheduling and priority queuing scheme named as (intra-ONU scheduling (*Ma, Zhu & Cheng, 2003*)). They are supposed to support delays in varied sensitive services. Dynamic

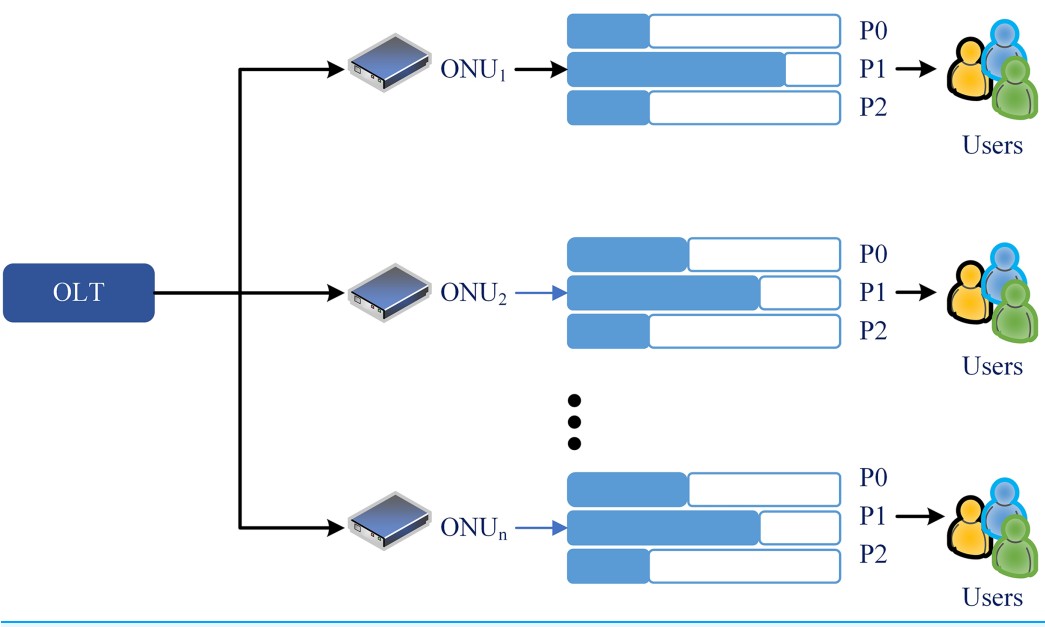

**Figure 14 Intra-ONU scheduling.**

Bandwidth allocation can be considered in the internal and external ONU scheduling as shown in Fig. 14. The OLT always deals with high-priority messages first, conferring strict priority scheduling so that the OLT ignores the QoS guarantee of the low priority, leading to increased packet delay and packet drop rate (*Luo & Ansari, 2005*).

To sum up, the benefit of this method is that the downlink channel grants the bandwidth to the ONUs as per the service level agreement. On the other hand, upstream link transmission has one downside: it grants static bandwidth that reduces the bandwidth and the throughput of uplink transmission.

### Efficient DBA

Based on the MPCP mechanism, this algorithm assigned the extra bandwidth of lightly loaded ONU to heavily loaded ONU. In this way, it significantly improved the network's performance, minimised the packet delay and length of the queue, and increased the throughput of heavily loaded traffic. To enhance bandwidth efficiency during periods of heavy network traffic, an operational scheduling control system is implemented to identify and report instances of idle time (*Zheng, 2006*).

### IPACT with grant estimation DBA algorithm

This algorithm is a development of IPACT. It guesses the recently arrived packets between two successive polling and granting ONUs sideways with the further assessed amount. In small traffic situations, the data packets communicate the consecutive cycle, thus dropping the waiting delay (*Zheng, 2006*).

### Two-layer bandwidth allocation DBA algorithm

This algorithm works in a double layer, as the name suggests. It divides the transmission into two parts: allocate bandwidth to upstream works in the primary layer and, in second,

**Table 3 Comparison of EPON DBA algorithms.**

| EPON DBA | Key strengths | Key weaknesses | Quantitative insights |
|---|---|---|---|
| IPACT (*Kramer, Mukherjee & Pesavento, 2002b*) | Simple, adaptive cycle time reduces idle time. | No QoS support, polling overhead scales with ONU count. | Delay: 12 ms for 32 ONUs; idle time reduced by 15–20%. |
| DBA for multimedia (*Choi & Huh, 2002*) | QoS-aware, prioritizes multimedia traffic. | High complexity, limited scalability, and priority starvation. | The latency for high-priority traffic was reduced by 30%, but the delay of low-priority traffic increased to 25 ms for 64 ONUs. |
| BGP (*Ma, Zhu & Cheng, 2003*) | SLA compliance, fairness in bandwidth allocation. | Polling overhead, inefficiency in unused slot reassignment. | SLA compliance: 99% for 16 ONUs; throughput dropped by 8% with 64 ONUs. |
| IPACT-GE (*Zhu & Ma, 2008*) | Reduces idle time and is better for bursty traffic. | High complexity, relies on accurate estimation. | Latency was reduced by 40% compared to IPACT for 16 ONUs; bandwidth under-utilization increased by 12% with errors. |

allocate the bandwidth to the whole ONU. Below the high traffic load, it assures the lowest bandwidth (*Choi & Park, 2010*). The comparison of all DBAs in this subsection is given in Table 3.

## DBA algorithm for GPON

The prior PON used TDM or a predefined pattern to distribute bandwidth to the upstream channel. GPON transmits uplink communications *via* DBA. The OLT may allocate bandwidth per-ONT or per-ONT-per-service (T-CONT) utilizing DBA approaches. The basic DBAs to help the understanding include Status Reporting DBA (SR-DBA), Non-Status Reporting DBA (NSR-DBA), and Hybrid DBA. SR-DBA uses ONU buffer occupancy reports to the OLT to allocate bandwidth dynamically to real-time demand, although it adds reporting overhead. Instead of ONU reporting, NSR-DBA has the OLT estimate bandwidth requirements based on previous data or specified thresholds, saving overhead but frequently resulting in less accurate allocations (*Feknous et al., 2015*). Hybrid DBA balances accuracy and overhead by employing ONU status reports and estimate methodologies (*Ozimkiewicz et al., 2010*). The key DBAs are presented below:

### GIANT DBA algorithm

The first DBA algorithm for GPON is Giga PON access network (GIANT). While distributing the bandwidth as per TCONT priority, the scheduling is done round-robin. When bandwidth is distributed to a queue, it only uses a down counter for every frame duration and is down by one for each frame duration. Upon expiration, the increased idle time generates bandwidth allocation for each service class, while some unused bandwidth remains (*Leligou et al., 2006*).

### Immediate allocation with colorless grant DBA algorithm

This algorithm is proposed to improve GIANT by minimising idle time and utilizing the unallocated bandwidth. In each ONU, a Byte counter (VB) is issued for immediate bandwidth allocation. When the down counter expires, VB needs to be recharged again by expiry; unallocated bandwidth also gets wasted, and ONU works through TCONT5 (*Han et al., 2008*).

**Table 4 Comparison of GPON DBA algorithms.**

| GPON DBA | Key strengths | Key weaknesses | Quantitative insights |
|---|---|---|---|
| GIANT (*Leligou et al., 2006*) | Simple and FSAN-compliant. Reliable for predictable traffic patterns. | High latency under dynamic traffic. Poor utilization for bursty traffic. Inefficient for high-speed modern networks. | Performs well for traffic loads <50%. Latency increases exponentially beyond 50% load. Utilization drops to 60% under bursty traffic. |
| IACG (*Sales, Segarra & Prat, 2014*) | Low latency for real-time applications. Dynamically reallocates unused bandwidth. Supports mixed traffic scenarios effectively. | Higher computational overhead. Susceptible to fairness issues. Less scalable for large ONU deployments. | 30% reduction in latency compared to GIANT. Utilization reaches 80–90% for mixed traffic. Fairness issues under high ONU contention. |
| GREAL (*Sales, Segarra & Prat, 2014*) | Optimized for long-reach GPONs. Reduces idle time and improves utilization. Significant latency reduction for long-reach systems. | Limited to long-reach applications. High downstream frame overhead. Increased complexity in long-reach synchronization. | 30% improvement in utilization for long-reach systems. Idle time reduced by 25–40%. Latency reduced by 20% in long-reach GPONs. |

### GPON redundancy eraser algorithm for long reach DBA

This algorithm overcomes long-distance networks' issues if the sent report is subtracted from the previous grant and leads to a real updated report form. At loads of lower traffic, bandwidth wastage is greater because of additional requests of high-priority users (*Sales, Segarra & Prat, 2014*). The comparison of all DBAs in this subsection is given in Table 4.

**Challenges at this stage**: GPON and EPON were successful, but they had major issues that prompted XG-PON and 10G-EPON. GPON's 2.5/1.25 Gbps bandwidth and EPON's 1 Gbps symmetric bandwidth were insufficient for 4K video, cloud services, and AR/VR. Static bandwidth allocation and ineffective DBA methods caused real-time service latency. Scalability issues hindered both protocols, and limited ONU support restricted urban installations. The lack of sophisticated QoS differentiation makes prioritizing latency-sensitive or high-priority traffic challenging.

## DBA algorithm for 10G-EPON

In a 10G-EPON, the MPCP mechanism remains unchanged. However, when considering the scheduling of the US channel, it is necessary to consider two equally cross-dependent EPON systems that utilize a single US channel. Consequently, there is a need to enhance the bandwidth allocation algorithms to schedule transmissions in this double-layered architecture. This involves considering the QoS parameters and improving the channel's efficiency and fairness.

### Advanced sleep-aware (ASDBA)

This algorithm exploits ONU's energy efficacy and DS and US scheduling, both done at the same transmission slot with its sleep mode. The ASDBA modifies the typical sequence of control message exchange used in the 10G-EPON system to adjust the time an ONU waits between sending a REPORT message and responding to a GATE message by converting it into the ONU's sleep time transmission. This feature allows the ONU to enter a state of continuous sleep after transmitting a REPORT message, remaining inactive until the start of the next transmission slot. This enhances energy efficiency in the ONU but also leads to delays in frame queuing (*Van et al., 2014*).

### DBA-gated algorithm

This is the first algorithm projected. It works with a very simple scheduling method. It grants the window size according to the demand of ONU without any composite calculation. It grants at first request maximum according to the demand of 50,000 bytes, and if asked for further bytes, ONU must resend the request of GRANT the window (*Rout, 2016a*).

### DBA-linear algorithm

As the name suggests, this algorithm grants window size to the user with a linear factor more than the demand; considering that may be due to delay, more bytes could be used. This shows that this algorithm is proportional to the requesting and grant window sizes. The window is assigned to ONU until the full data is transferred. In this regard, this algorithm offers good QoS (*Rout, 2016a*).

### DBA-MAX algorithm

In this algorithm, a threshold level is fixed for the GRANT window size for every user; if any user needs more windows than the threshold, it provides more to that user. Other than that, a fixed threshold value is assigned to every user. The advantage of this scheme is that other users will block no OLT, but the drawback is that GRANT window size is independent of REQUEST size, generating too much overhead (*Rout, 2016a*).

### TD-optimum algorithm

This algorithm eliminates the downside of the above DBA GATE algorithm; if the throughput and delay of the system vary, this algorithm can monitor the performance of the network well. It supports delay-sensitive traffic like voice and video over IP and non-delay sensitive traffic like massive data and video transfer applications, so it is the triple play. So, it did not drop data, voice, and video packets and communicate well (*Rout, 2016b*).

### Sub-MOS-IPACT DBA algorithm

At different granularities, this scheme promises bandwidth, individual ONUs, multi-ONU clients, and subgroups of ONUs. A Sub-group is a set of ONUs belonging to the same network. Customer multi-ONU bandwidth was also prioritized by sub-MOS-IPACT distribution of the same multi-ONU client among subgroups. The simulation results indicate that the proposed mechanism enhances the efficiency of the network (*Ciceri, Astudillo & da Fonseca, 2019*). The comparison of all DBAs in this subsection is given in Table 5.

## DBA algorithm for XG PON
### Immediate allocation with reallocation DBA algorithm

In 2012, the immediate allocation with reallocation (IAR) algorithm was introduced to overcome the unused bandwidth issue. In this algorithm, polling is introduced iteratively. Scheduling is achieved in four steps: First, from GPA to TCONT2 and TCONT3. Second, scheduling will repeat for overloaded queues for TCONT2 and TCONT3 so that

**Table 5 Comparison of 10G-EPON DBA algorithms.**

| 10G-EPON DBAs | Key strengths | Key weaknesses | Quantitative insights |
|---|---|---|---|
| ASDBA (*Van et al., 2014*) | Energy-efficient, reduces ONU power consumption by enabling sleep modes during idle periods. | Increased latency due to ONU wake-up times, unsuitable for delay-sensitive applications. | Reduced energy consumption by up to 30%, latency increased by 5–10 ms during high traffic loads. |
| DBA-Gated (*Rout, 2016a*) | Simplicity and predictable performance; ensure fairness through fixed grant allocation. | Bandwidth underutilisation when ONUs request less than their assigned quota. | Maintains 90% efficiency under steady traffic, drops to 70% with bursty traffic. |
| DBA-Linear (*Rout, 2016a*) | Dynamically adjusts bandwidth allocation, improving utilization for dynamic traffic patterns. | High computational overhead, less effective with sudden traffic spikes. | Achieved 95% bandwidth utilization, with 25% higher CPU load than DBA-Gated. |
| DBA-MAX (*Rout, 2016a*) | Redistributes unused bandwidth, ensuring high utilization and fairness among active ONUs. | Slight delays under heavy congestion due to real-time adjustments. | Provided 98% utilization, added an average latency of 10 ms during peak traffic. |
| TD-optimum (*Rout, 2016b*) | Optimized for delay-sensitive applications with traffic demand and delay thresholds. | High computational complexity and limited scalability for large ONU networks. | Reduced latency to 1–2 ms for high-priority traffic, increased processing power by 30%. |
| Sub-MOS-IPACT (*Ciceri, Astudillo & da Fonseca, 2019*) | Prioritizes services with high QoS demands, and offers multi-level bandwidth allocation for diverse needs. | High complexity in multi-level allocation, challenging real-time operation for traffic spikes. | Improved utilization by 15% over IPACT, QoS satisfaction rates of 98%, added delays of 10–12 ms for 50+ ONUs. |

under-load queues get bandwidth. Third, SPA to TCONT3 and TCONT4. Finally, TCONT5 gets allocation from CG (*Han, 2012*).

### Efficient bandwidth utilization

As repeating the scheduling process results in a delay to some level, this algorithm proposed that if we use the Borrow and Refund (BR) concept while updating the operations, this issue could be resolved. If the request is more than VB, it gets negative results in over-allocation. The unused bandwidth will compensate for this over-allocation in the previous queue. But this method makes low priority class get fewer assignments (*Han, Yoo & Lee, 2013*).

### Simple and feasible (SF) DBA algorithm

This algorithm proposed a simple method to utilize the unused bandwidth compared to the BR operation. The author proposed a service class using a down counter and a single available 448-byte counter and allocated the bandwidth at request to every queue. By this method, each service class ignores the unused bandwidth, but the limitation is one service class; it cannot use the bandwidth of other service classes to avoid depletion (*Han, 2013*).

### DBA with high utilization (DBAHU) algorithm

The updated version of SFDBA is DBAHU. This algorithm used BR in the updated operation of SFDBA. This can fix the problems in the above algorithm. But it will disturb the next cycle if there is no unused bandwidth. As round-trip time (RTT) is large in long-distance PON, this OLT will receive an extra report and waste the bandwidth (*Han, 2014*).

### Improved bandwidth utilization DBA algorithm

It improves the scheduling and polling mechanism by assigning a slot three times to DBRu when the counter range is down to 2, 5, and 8. When it expires, TCONT4 starts execution. Because of the high priority of TCONT3 and TCONT2, it mostly focused on them. But the issue is that as TCON4 is executed after the down counter expires, this quickly increases the delay and ignores the unused bandwidth (*Butt et al., 2017*).

### IRIS DBA algorithm

After the execution of SPA and GPA, IRIS will stand by for the bandwidth phase. Standby bandwidth starts from the finish of the last allocation and flinches of the fresh grant cycle. It performs better in TCONT4 and TCONT3 than DBAHU. IRIS later used the standby bandwidth by predicting the queue size and allocating its percentage to the queue. It used only 8 ONUs to show the results (*Kyriakopoulos & Papadimitriou, 2016*).

### Comprehensive bandwidth assignment scheme for long reach DBA algorithm

Comprehensive bandwidth assignment scheme for long reach (CBA-LR) enhances the scheduling and polling mechanisms used in algorithms such as GREAL, EBU, and IACG by efficiently assigning residual bandwidth to the appropriate traffic class. The residual bandwidth can be utilised more effectively by eliminating the BR concept in the EBU. However, this comes at the cost of increased delay for traffic classes T2 and T3 compared to the EBU algorithm (*Butt et al., 2019*).

### Comprehensive bandwidth utilization DBA algorithm

It is better to overcome the downsides in polling and scheduling IACG and EBU by allocating the whole bandwidth by ONU and OLT to TCONT and avoiding bandwidth wastage. BR concept is not used such that extra bandwidth will be moved to TCONT solitary (*Butt et al., 2018b*).

### Sleep assistive DBA algorithm

It works over ONU on its sleep mode cycle to save energy because polling only happens on ONU during SI in odd cycles. The GPA is according to SLA limits for T4, T3, and T2, but it is assigned to active state ONUs when near RBW (*Butt et al., 2018a*).

### Demand forecasting (DFDBA) algorithm

Using statistical modeling, forecast the demand in future stores' last 100 demand cycles using a circular buffer. By this, a significant amount of delay gets reduced (*Memon et al., 2019*). The comparison of all DBAs in this subsection is given in Table 6.

**Challenges at this stage**: Comparing the different versions of DBAs from both standards, it is observed that the fixed designs and restricted wavelength support were unsuitable for quickly expanding traffic from 4K streaming and cloud services, limiting bandwidth scalability. Asymmetric upstream bandwidth limits real-time gaming, video conferencing, and 5G fronthaul. Latency and QoS control added to the challenges. Emerging applications like AR/VR and mission-critical 5G services need minimal latency and jitter, but DBA techniques were too inflexible. Fairness and

**Table 6 Comparison of XG-PON DBA algorithms.**

| XG-PON DBAs | Key strengths | Key weaknesses | Quantitative insights |
|---|---|---|---|
| IAR (*Han, 2012*) | Immediate allocation reduces latency and minimises idle bandwidth. | Frequent reallocations introduce computational overhead under high traffic loads. | Idle time reduced by up to 15%, latency increased by 10% in heavy traffic. |
| EBU (*Han, Yoo & Lee, 2013*) | Maximises bandwidth utilization by redistributing unused bandwidth efficiently. | Fairness can be compromised during peak demand, affecting low-priority traffic. | Utilization improved by 20%, delays for real-time traffic increased by 5%. |
| SF (*Han, 2013*) | Simple and lightweight, reducing computational complexity. | Inefficient in diverse and dynamic traffic patterns. | Complexity reduced by 30%, fairness degraded by 25% for heterogeneous traffic. |
| DBAHU (*Han, 2014*) | Balances high utilization and fairness among ONUs. | Increased computational demands limit scalability for large networks. | Utilization improved by 10%, latency increased by 8% under high traffic loads. |
| IBU (*Butt et al., 2017*) | Reduces idle time effectively and balances bandwidth allocation. | Performs poorly in handling highly bursty or dynamic traffic patterns. | Idle time reduced by 40%, latency increased by 5–7 ms. |
| IRIS (*Kyriakopoulos & Papadimitriou, 2016*) | Adapts to predictable traffic patterns using historical data. | Struggles with anomalies or highly volatile traffic conditions. | Latency reduced by 20% for predictable traffic, degraded by 15% under traffic bursts. |
| CBA-LR (*Butt et al., 2019*) | Efficient for large-scale long-reach deployments using advanced polling mechanisms. | High computational complexity adds processing delays. | Utilization improved by 30%, polling overhead increased by 10–15 ms. |
| CBU (*Butt et al., 2018b*) | Maximises fairness and utilization with efficient polling and allocation strategies. | Higher latency for real-time services due to complexity. | Fairness improved by 25%, delays for real-time traffic increased by 5%. |
| Sleep assistive (*Butt et al., 2018a*) | Reduces power consumption without significant performance degradation. | Latency increases, making it less effective for ultra-low latency applications. | Energy consumption reduced by 30%, latency increased by 5–10 ms. |
| DFDBA (*Memon et al., 2019*) | Uses traffic prediction to allocate bandwidth proactively, reducing latency. | Computationally intensive due to real-time forecasting models. | Latency reduced by 20–30%, computational overhead increased by 10%. |

resource allocation under bursty traffic remained unsolved, especially in different service settings. Energy efficiency was another issue. As the number of connected devices multiplied, both standards' high power consumption, especially at ONUs, became unsustainable. Scalability and flexibility issues prevented them from adapting to different traffic patterns and multi-service networks, which are essential for digital infrastructure.

## DBA algorithm for NG-EPON

In NG-EPON, there are four wavelength channels, and ONU shares upstream bandwidth in TDM form over separate wavelengths. It also works in a request-and-grant manner, depending on the control messages of Ethernet (REPORT and GATE). The OLT side will decide on individual ONU's bandwidth and wavelength time slot. In NG-EPON, DWBA handovers transmission wavelengths and time slots to an ONU dynamically at its request during upstream transmission to avoid any conflict.

**Table 7 Comparison of NG-EPON DBA algorithms.**

| DBA algorithm | Key strengths | Key weaknesses | Quantitative insights | Practical applicability |
|---|---|---|---|---|
| Fair DWBA (*Hussain, Hu & Li, 2017*) | Ensures fairness in bandwidth allocation; minimises resequencing problems. | Increased computational complexity for fairness optimization; not suitable for latency-sensitive applications. | Fair bandwidth allocation can improve throughput by up to 10% in bursty traffic scenarios. | Applicable for industrial PONs requiring fairness across multiple ONUs; less suited for 5G/6G fronthaul. |
| First-Fit (FF) DWBA (*Hussain et al., 2017*) | Low latency; simple to implement; improves resource utilization. | May lead to inefficient bandwidth allocation for diverse traffic types. | Latency reduced by 20% compared to static allocation methods in simulations with mixed traffic. | Suitable for 5G fronthaul with low-latency requirements but lacks advanced QoS capabilities for industrial environments. |
| Agile wavelength (AW) DWBA (*Wu, Wang & Guo, 2018*) | High upstream bandwidth utilization; manages heterogeneous propagation delays effectively. | Complexity in wavelength management; requires precise delay estimation. | Achieves >90% bandwidth utilization in scenarios with varying propagation delays. | Highly effective for 5G/6G fronthaul where heterogeneous propagation delays are a concern. |
| Flexible wavelength (FW) DBA (*Hussain et al., 2018*) | Adaptable to dynamic traffic patterns; improves bandwidth efficiency. | Complex implementation at the ONU level; higher control overhead. | Improves throughput by 15% in scenarios with dynamic traffic demands compared to static DBA. | Suitable for industrial PONs requiring adaptive bandwidth allocation and flexibility. |
| MUMS DBA (*Yan et al., 2018*) | Efficient handling of multi-transceiver ONUs and diverse services. | High-priority services may dominate bandwidth allocation, leading to fairness issues. | Reduces packet delay variation by 30% for high-priority services in multi-user scenarios. | Ideal for industrial PONs with diverse service classes but less suited for latency-critical 5G fronthaul. |
| QoS DWBA (*Rafiq & Hayat, 2019*) | Guarantees QoS for diverse traffic classes; prioritizes high-priority traffic effectively. | Limited bandwidth for low-priority traffic; underutilization in lightly loaded systems. | Latency for high-priority traffic reduced by 25% compared to non-QoS DBAs. | Critical for 5G fronthaul where QoS guarantees are essential for eMBB and URLLC traffic. |
| SCAP DWBA (*Wang, Guo & Hu, 2018*) | Reduces reordering issues; ensures higher throughput. | Potential underutilization of other channels due to the "single channel" constraint. | Improves throughput by 18% compared to FIFO-based DBAs in high-load scenarios. | Effective for 5G fronthaul applications with specific channel constraints but less adaptive to bursty industrial traffic. |
| FIFO DBA (*Raad, Inaty & Maier, 2019*) | Simple and effective for reducing latency; easy to implement. | Struggles with diverse traffic types and lacks prioritization mechanisms. | Latency reduction of 15% for single-class traffic compared to basic polling methods. | Suited for straightforward 5G fronthaul use cases but not robust enough for industrial PONs requiring complex QoS handling. |
| Reservation pattern (RP) DBA (*Raad, Inaty & Maier, 2019*) | Improves throughput and latency for predictable traffic patterns. | Limited adaptability for bursty or unpredictable traffic patterns. | Optimizes throughput and latency for predictable traffic patterns by up to 15%. | Ideal for industrial PONs and environments with steady traffic patterns; less effective in dynamic scenarios. |
| Deep learning-based DBA (*Hatem, Dhaini & Elbassuoni, 2019*) | Highly adaptive to traffic variations; reduces control overhead; future-proof for dynamic environments. | Computationally expensive; requires large datasets for training; real-time inference may be challenging. | Predicts traffic patterns with 95% accuracy, reducing packet delay by up to 30% in simulations. | Ideal for 5G/6G fronthaul with dynamic and bursty traffic; potential for industrial PONs with AI-driven adaptability needs. |

| Table 7 (continued) | | | | |
|---|---|---|---|---|
| DBA algorithm | Key strengths | Key weaknesses | Quantitative insights | Practical applicability |
| Reinforcement learning-based DBA (*Zhou et al., 2020*) | Optimizes bandwidth allocation over time; adapts to network conditions dynamically; improves QoS and latency. | High computational overhead; requires training time to achieve optimal performance. | Latency reduced by 35% and throughput increased by 20% compared to static DBA in complex traffic scenarios. | Highly suitable for 5G/6G fronthaul and industrial PONs requiring self-adaptive and intelligent resource allocation. |
| Mobile CPRI predictive (MCP) (*Araujo et al., 2019*) | Optimized for CPRI traffic; reduces latency through advanced prediction. | Requires accurate and real-time prediction models; high computational complexity. | Reduces latency by 40% compared to basic DWBA in mobile fronthaul applications. | Specifically designed for 5G/6G fronthaul scenarios with CPRI traffic but not generalizable for diverse industrial PONs. |
| Efficient bandwidth management (EBMA) (*Rafiq & Hayat, 2020*) | Maximises resource utilization; handles diverse traffic scenarios effectively. | Higher complexity for managing bandwidth dynamically across multiple ONUs. | Achieves up to 95% resource utilization in simulations with heterogeneous traffic patterns. | Suitable for both 5G/6G fronthaul and industrial PONs with diverse traffic and high utilization requirements. |

### Fair DWBA algorithm

*Hussain, Hu & Li (2017)* proposed a fair algorithm to reduce the re-sequencing problem of frames that happened when a single grant ONU transmits data on a single wavelength. Controlling this issue increased the fairness of scheduling. It improved bandwidth usage efficiency by assigning the allocated bandwidth, thus minimising bandwidth wastage. In general, this results in a minimal packet delay and loss ratio for all levels of network traffic. The comparison of all DBAs in this subsection is given in Table 7.

### DWBA algorithm

In addition to the fair algorithm, this first-fit (FF) algorithm also minimises the transmission latency along with the mitigation of the re-sequencing problem, so by increasing the efficiency of bandwidth by decreasing the disproportionate use of guard time, it also reduced the wastage on a single wavelength, just like fair scheme do in results, giving low packet delay (*Hussain et al., 2017*).

### Agile wavelength (AW) DWBA algorithm

This scheme schedules the ONU in ascending order of their RTT and tries to grant ONU on a single wavelength channel to avoid the guard time from waste of excessive bandwidth. This also adjusts some ONU on multiple wavelength channels and ensures the transmission is finished simultaneously. This scenario maximised the upstream bandwidth of channels, which is the best-proposed scenario for better utilization of bandwidths. This also lowered the average delay and satisfied the high-load ONUs (*Wu, Wang & Guo, 2018*).

### Flexible wavelength DBA algorithm

This scheme reduces the transmission delay. It grants bandwidth to ONU according to load traffic and is flexible in scheduling. A high traffic load assigns more bandwidth to those ONUs and vice versa. Bandwidth allocation also depends on the number of total ONUs in the network and the mitigation of re-sequencing delay for light-loaded ONUs. It

also does this for heavy-load ONUs. It also introduced a parameter alpha by which flexible bandwidths are decided. It also depends on the number of ONUs in a network (*Hussain et al., 2018*).

### Multi-type users and multi-type services DBA algorithm

This algorithm is proposed to facilitate residential and business users at the same OLT and supports QoS parameters. It comprises numerous sub-algorithms like WoF and ToF grant sizing and scheduling to resolve the sub-problems of the transceiver ONUs DBA synthetically. Firstly, a two-dimensional priority multi-type users and multi-type services (MUMS) queue model is designed to determine grant scheduling among MUMS. Later, grant sizing is introduced, and further ToF and WoF too. It also reduced the average packet delay and packet loss ratio as well as the jitter delay (*Yan et al., 2018*).

### Quality of service DWBA algorithm

This scheme can fulfil the demands of network users in the future. This QoS-based algorithm handles LP (low priority) and HP (high priority) traffic simultaneously. It minimises the delay that ONU faces in the online manner. The HP traffic case for LP works offline because it does not need a constant bit rate and can bear delay in data transmission (*Rafiq & Hayat, 2019*).

### Single channel as possible DWBA algorithm

The single channel as possible (SCAP) algorithm reduces frame reordering issues, which occur simultaneously with parallel transmissions. If among multiple wavelengths, the grant distributes without considering grant size, and the ONU transmits data among all channels, resulting in frame reordering. This algorithm controls this issue by granting bandwidth in single channels. Along with this, packet delay also improved (*Wang, Guo & Hu, 2018*).

### First in first out DBA algorithm

The network capacity in the NG-EPON standard increased by 100 GB/s, and strict requirements must be applied to delay packets. New scheduling criteria were introduced, and the grant assigned upstream is now CDMA-based on assuring the meetup of the new required network throughput. This algorithm improved network capacity by reducing the mean waiting time for received packets and jitter delay. In first in first out (FIFO), ONU ordered the request according to the arrival time of requests (*Raad, Inaty & Maier, 2019*).

### Reservation pattern DBA algorithm

This is also used for CDMA-based NG-EPON networks. It differed from FIFO in the scheduling sense that ONU transmits the queue according to the predefined reserved pattern. Its objective is the same as that of FIFO (*Raad, Inaty & Maier, 2019*).

### Deep learning DBA

This DBA analyzes previous use patterns and real-time network metrics to estimate ONU traffic demands using deep neural networks. The model allocates bandwidth based on expected demand, guaranteeing effective resource distribution. Training the system on big

datasets allows it to dynamically adapt to changing traffic scenarios, decreasing packet delays and optimizing network resources (*Hatem, Dhaini & Elbassuoni, 2019*).

### Reinforcemnt leaning based DBA

This DBA uses reinforcement learning to dynamically adjust bandwidth allocations by interacting with the network environment. It learns optimal strategies based on feedback, ensuring effective bandwidth distribution in unpredictable situations and continuous improvement through experimentation (*Zhou et al., 2020*).

### Mobile CPRI predictive (MCP)-DWBA algorithm

It predicts how many time slots ONU will need in the future, after which there will be no need for grant messages. In this way, it saves the propagation time spent on grant messages so that time can now be used to transmit data. Its objective is the same as that of the MC-DWDBA algorithm. The MCPDWBA method optimizes the utilization of time slots, decreases the number of active channels, and increases the pace at which data is transmitted, hence facilitating the deployment of NG-EPON as the MF networks (*Araujo et al., 2019*).

### Efficient bandwidth management DWDBA algorithm

It is designed to bring about expanded traffic supplies for changed ONUs. EBMA routine is estimated moderately with adapted first fit (FF-DWBA) and IPACT (M-IPACT) algorithms. It is better in packet delay and drop ratio; it segregates both low-loaded (LL) and heavy-loaded (HL) users (*Rafiq & Hayat, 2020*). The comparison of all DBAs in this subsection is given in Table 7.

## EVOLVING PON APPLICATIONS AND KEY DBA CHALLENGES

This section describes the potential emerging applications of PONs, including 5G fronthaul networks, FTTR, and industrial PONs, which require major efforts and research to standardize. Each application is overviewed in terms of its current state of the art, and challenges and opportunities are described in each subsection.

### TDM PON based 5G fronthaul network

Intelligent technologies have significantly improved living convenience in the last decade. Cisco predicts that interconnected devices in cellular networks will reach billions by 2030 due to IoT, vehicular platooning, and immersive virtual reality (VR) applications (*Report, 2023*). 5G technology provides high data transfer rates, little latency, and dependable device performance. 5G networks are composed of a significantly larger number of high-capacity cells in the access layer, which poses the difficulty of interconnecting these cells to the remainder of the network. To achieve this objective, it is essential to implement a high-capacity transport connection for each of these tiny cells. Unlike the 4G backhaul network (which refers to the link between the base station and the core network), 5G radio access network (RAN) will employ several transport network topologies as shown in Fig. 15. The main design idea for the 5G access layer is the cloud/centralized RAN

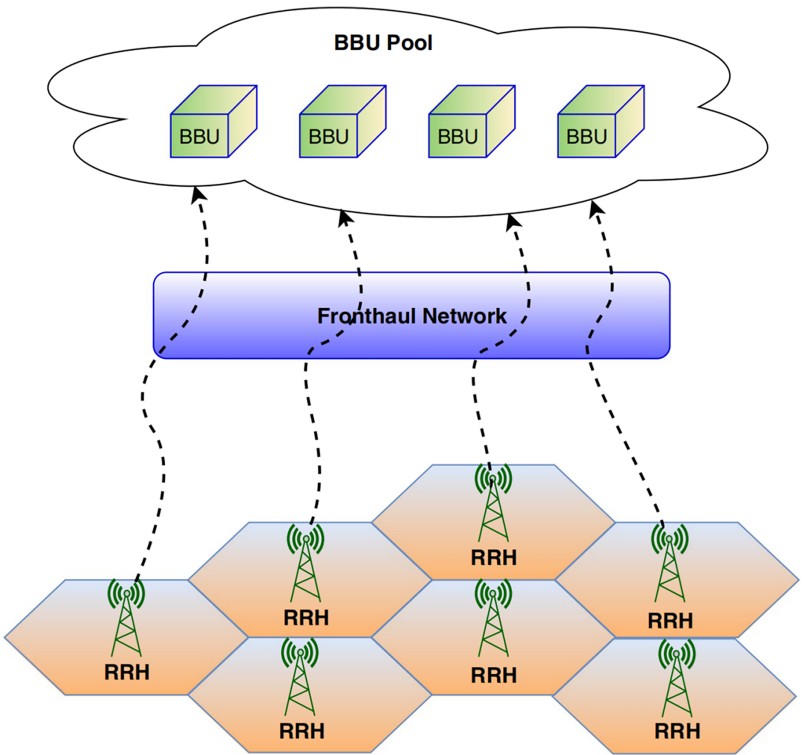

**Figure 15  5G fronthaul architecture.**

(C-RAN). This design puts all the processing of wireless signals in one base band unit (BBU) at a central location (*Jaffer et al., 2020*). A pair of antennas are the only components installed at the cell site in this architecture, known as the remote radio head (RRH) or radio unit (RU). Leaving only antennas at the RRH is just one of many different fronthaul split point options. At the same time, a group of BBUs is usually grouped at a central office (CO). The BBU and RRH/RU connection is known as the fronthaul link. A fronthaul network in basic C-RAN uses the Common Public Radio Interface (CPRI) protocol to transmit data between RRH and BBU, and it requires optical links. CPRI is a digital RoF technique that transmits the IQ data of the baseband transmissions. CPRI necessitates a PtP fiber connection due to its need for higher bandwidth.

However, the deployment of PtP fiber links at every cell site is very costly or impossible for mobile network operators. Therefore, the optimized positioning of the elements in C-RAN architecture is necessary to deploy a cost-effective network. The researchers have presented various new network dimensioning approaches for fronthaul, as discussed in *Ranaweera et al. (2018)*. This article examines the physical-layer split option method for implementing a 5G fronthaul network. In the physical-layer split option, the various functions of the BBU are relocated at the RRH to minimise the data rate or the requirement for the PtP fiber link at the fronthaul interface. However, there is still a need for a high-capacity fronthaul network with low latency to implement basic C-RAN for 5G.

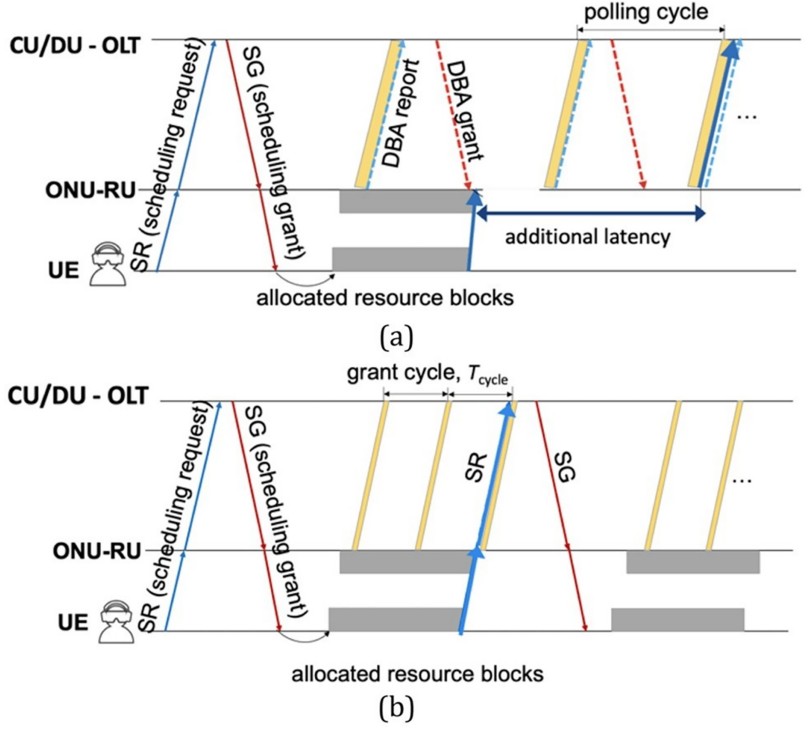

**Figure 16 (A) Conventional and (B) cooperative DBAs (*Wong & Ruan, 2023*).**

Several investigations suggest that the deployment of TDM-PON-based mobile fronthaul (MFH) is one of the most prominent fronthaul solutions for the physical layer split option that meets the 5G network and beyond requirements, especially of ultra-low latency 5G application, *i.e.*, up to 1 millisecond (ms). Nevertheless, these architectural adjustments alone are inadequate for attaining latency as low as 1 ms since it also necessitates an effective allocation of the available bandwidth resources. One can employ an effective resource allocation approach known as the DBA scheme for TDM PON-based CRAN architecture to achieve this.

### DBA algorithms for 5G TDMA PON based fronthaul traffic and challenges

The 5G fronthaul network can support many types of traffic, notably high-priority traffic for real-time services like audio and video and lower-priority traffic for data exchanges. NG-MFH networks need to handle many different services and types of traffic. This makes it even more important to have a DBA that can handle traffic demand, quality of service needs, and service level agreements (SLAs) for 5G customers. Implementing PON technology for MFH solutions and resource sharing in convergent networks poses challenges. TDM involves sharing US bandwidth, leading to contention for uplink capacity and collisions in the RAN payload. Allocating the appropriate bandwidth and determining the transmission timing is critical in the US to prevent congestion and fulfil latency requirements. RAN scheduling involves the user equipment (UE) requesting transmission by submitting a scheduling request (SR) to the centralized/distributed unit (CU/DU). The

CU/DU subsequently plans and allocates radio resources by returning a scheduling grant (SG) to the UE. In Ethernet PONs, the standard baseline DBA (*Kramer, Mukherjee & Pesavento, 2002b*) uses report and grant messages to ask for and give bandwidth to the MFH, which causes poor latency. The extra delay is shown in the timing diagram in Fig. 16A. The cooperative DBA approaches utilize fronthaul traffic data *via* a cooperative transport interface message to tackle this problem (*Tashiro et al., 2014*). The CU/DU collaborates with the OLT by sharing previous knowledge about UE traffic, which is determined by analyzing how packets arrive, *i.e.*, traffic patterns at the ONU-RU. If one looks at CPRI traffic with a fixed bit rate and packet size, as well as the patterns of packet arrival at ONU-RUs or the inter-packet gaps (IPGs) in MFH transmissions, one can find out how much bandwidth each ONU-RU is given. This is called $T_{grant}$. The interval between two successive transmissions by an ONU RU is known as a grant cycle, denoted as $T_{cycle}$. By coordinating the transfer of $T_{grant}$ with SG, the grant-report procedure of the standard baseline DBA can be removed, as shown in Fig. 16B.

*Wong & Ruan (2023)* and *Zhang et al. (2018)* presented ML-driven and Cell-Average Traffic (CAT)-based DBA techniques to enhance bandwidth allocation and maintain QoS in MFH networks. The former DBA made a fast and self-adaptive bandwidth (FSA) decision in response to traffic patterns, network loads, and line rates. This enhanced the DBA scheme's ability to handle various applications in future MFH networks. The latter DBWA approach uses CAT to guarantee QoS for different types of cells in MFH. The work of *Lagkas et al. (2021)* described a complete method for allocating radio, optical, and MEC resources together. This method uses advanced technologies to make resource allocation dynamic and energy-efficient. *Kalfas et al. (2019)* discussed the unique constraints of 5G mmWave networks and suggested a fiber wireless fronthaul design with a DBA protocol to address capacity and latency needs. All these studies emphasize the significance of DBA in NG-MFH networks and the ability of several methods to meet this requirement.

In industries that host DOCSIS, PON, and 5G systems, a converged environment needs unique and interoperable (AI/ML) DBA design solutions (*Poletti, 2024*), even though it reduces uplink traffic (*Garima, Jha & Singh, 2024*) and gives too much fronthaul support (*Feng et al., 2023*). Keeping AI/ML developments in focus for the two different optical and wireless domains and self-optimizing networks that adapt to link access and network requirements can increase the efficiency of resource sharing. It is expected that 6G would use SDN's current enabling technologies to improve networks by attempting to render them more modular, accessible, and self-driving. Table 8 presents the challenges and opportunities in DBAs for 5G/6G fronthaul networks.

## FTTR

FTTR brings a PON access network inside a home/building/hotel in place of a single ONU to satisfy the high bandwidth requirements of its tenants. FTTR design employs two layered PON to link residential customers (see Fig. 17). One layer is the access PON, which employs a commercial PON system for FTTH, while the other is the FTTR home network (FHN). The main FTTR unit (MFU) joins multiple sub-FTTR units (SFUs) to construct an FHN with fiber as an internal backbone. MFU with dual functionalities acts as an ONU

**Table 8 Challenges and opportunities in DBA for 5G/6G TDMA PON based fronthaul.**

| Challenge | Opportunity | Actionable step | Target |
|---|---|---|---|
| Handling dynamic traffic in 5G fronthaul (*Cao et al., 2021*; *Zaouga et al., 2021*; *Fayad, Cinkler & Rak, 2024*) | Develop adaptive and self-adjusting DBA strategies that dynamically adjust to real-time network changes without retraining models. Integrate AI-driven anomaly detection. | Implement AI-driven anomaly detection models to enhance adaptation in DBA. | Maintain low latency (<250 $\lambda$s) and minimise prediction error (<10%) even under fluctuating fronthaul demand. |
| Predictive accuracy in high traffic loads (*Cao et al., 2021*; *Ciceri et al., 2024*) | Improve predictive DBA with reinforcement learning (*e.g.*, fine-tuning model parameters using feedback loops). | Use reinforcement learning for adaptive bandwidth prediction based on previous traffic trends. | Ensure delay improvement of at least 10% in high-load scenarios ($\geq$0.5 network load). |
| Balancing 5G and best-effort services (*Zaouga et al., 2021*; *Rawshan, Hossen & Islam, 2024*) | Develop priority-aware DBA that dynamically adjusts allocation between services while maximising total throughput. | Implement traffic prioritization techniques to ensure fair resource allocation. | Achieve high resource utilization ($\geq$%) while ensuring 5G latency compliance. |
| Traffic dynamics in APP layer (*Hu et al., 2023*; *Garima, Jha & Singh, 2024*) | Develop joint MFH-MEC coordination to allocate resources based on real-time APP-layer traffic predictions dynamically. | Combine DBA and MEC scheduling into a single optimization framework. | Reduce latency violation ratio (LVR) by 30% and Achieve MAPE prediction error <10% for upcoming traffic forecasts. |
| High network delay and algorithm complexity of DLMs (*Du et al., 2023*; *Murphy, Townsend & Antony, 2024*) | Use extreme learning Machine (ELM)'s fast learning speed, which achieves 600× faster training than LSTM/GRU while maintaining prediction accuracy. | Reduce computational overheads, possibly. | Train in 0.03s (ELM) (or even less) *vs.* 18s (GRU) and 29s (LSTM). |

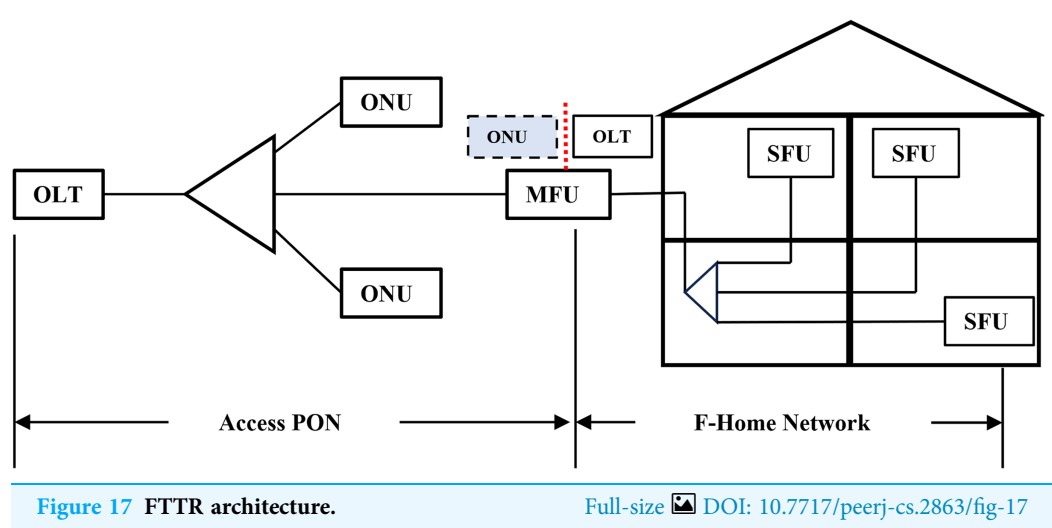

**Figure 17 FTTR architecture.**

and OLT to communicate with the PON network and SFUs, respectively. The established FTTR features in ITU-T Recommendation G.9940/41 cover optical link and Wi-Fi coordination, optical transmission requirements, and average data speeds. In late 2023, power budget, frame order, and control message parameters were decided to encourage the different application cases and deployment. Over 1 million people have used FTTR's gigabit/s home networking services in the previous 3 years, with operators and

**Table 9 Challenges and opportunities in DBA for FTTR applications.**

| Challenge | Opportunity | Actionable step | Target |
|---|---|---|---|
| Traffic management in multi-room environments and handling traffic bursts (*Cases, 2021*; *Philpot & Wilson, 2022*; *Wu et al., 2023*) | Ensure stable connectivity with seamless roaming and low latency even during fluctuating traffic loads. | Implement roaming-aware and predictive DBA algorithms to allocate bandwidth dynamically based on mobility and historical traffic patterns. | Reduce roaming latency to <10 ms and ensure >95% service reliability during traffic bursts. |
| Bandwidth allocation for gigabit-level services and integration with Wi-Fi and optical systems (*Philpot & Wilson, 2022*; *Zan et al., 2024*; *Wei et al., 2024*) | Optimize bandwidth allocation for high-speed applications while ensuring unified management of Wi-Fi and optical components. | Develop hybrid and application-aware DBA techniques to prioritize latency-sensitive applications and integrate wireless and fiber networks seamlessly. | Maintain latency <50 ms for AR/VR and telemedicine services with 99% utilization of network resources. |
| Energy consumption for 24/7 FTTR operations (*Philpot & Wilson, 2022*; *Cai et al., 2024*) | Reduce energy usage while maintaining high bandwidth availability. | Implement energy-efficient DBA algorithms that activate power-saving modes during low-traffic periods. | Decrease FTTR energy consumption by 15–20% without service degradation. |
| Standardization and interoperability (*Cases, 2021*; *Philpot & Wilson, 2022*; *Zhang et al., 2024*) | Drive industry adoption through standardization of DBA for FTTR. | Collaborate with ITU-T to establish standard DBA protocols for unified management across vendors. | Enable interoperability for 90% of devices by 2026. |

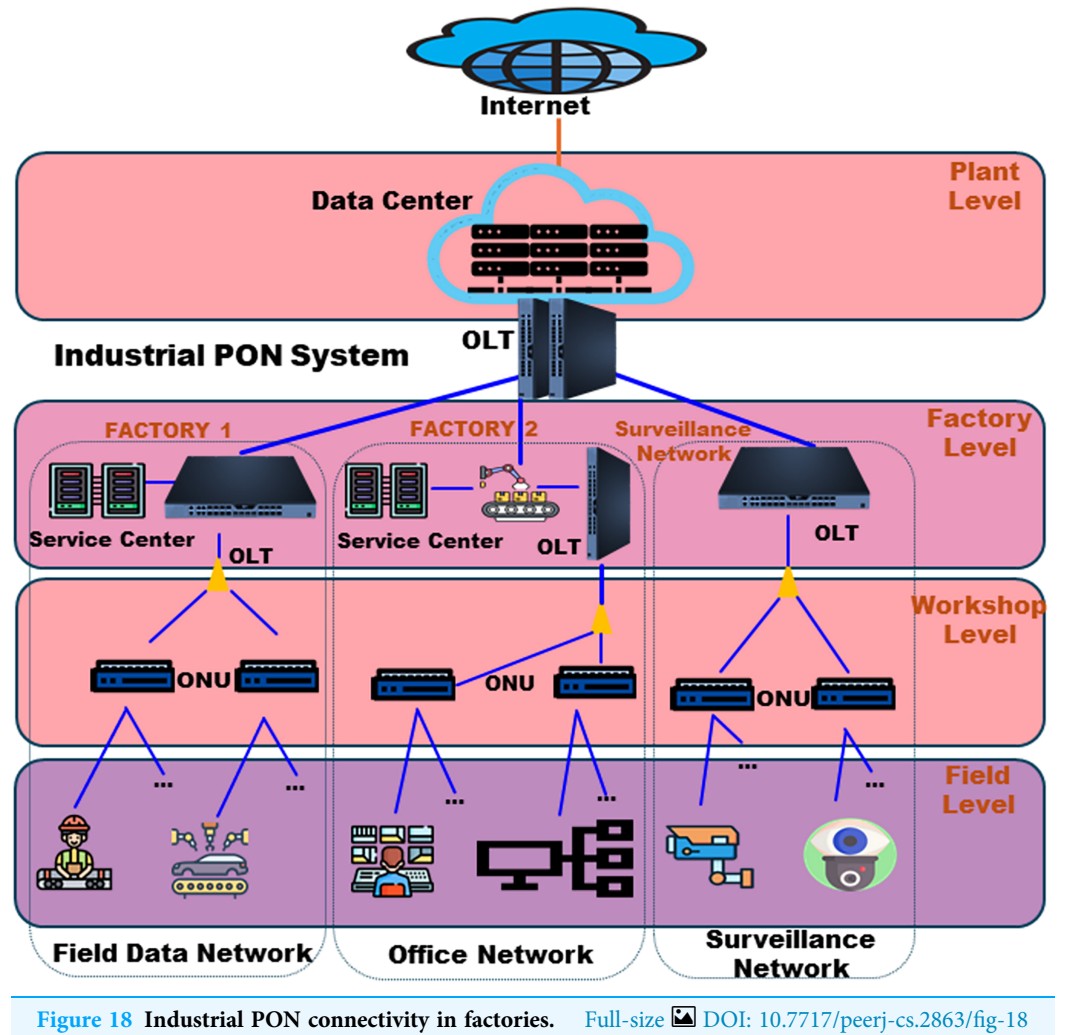

**Figure 18 Industrial PON connectivity in factories.**

**Table 10 Challenges and opportunities in DBA for industrial PONs.**

| Challenge | Opportunity | Actionable step | Target |
|---|---|---|---|
| Real-time communication and synchronization for industrial operations (*Luo et al., 2023*; *ITU-T, 2024a*) | Enable reliable, ultra-low-latency communication and precise synchronization for mission-critical industrial processes. | Investigate low-latency protocols, implement deterministic control for bounded latency, reduce jitter to <1 ms, and incorporate effective scheduling algorithms. | Achieve latency ×, jitter <100 ≥1 mss, and precise real-time communication for control systems. |
| Seamless scalability and connectivity for industrial IoT devices (*Jiang et al., 2023*; *Rayhan, Natali & Apriono, 2024*) | Support a growing number of IIoT devices, sensors, and industrial equipment with seamless protocol interoperability. | Design hierarchical DBA schemes, address synchronization, TSN, and QoS mechanisms for dynamic bandwidth allocation. | Support 10,000+ devices, ensure 100% QoS compliance, and provide protocol-level interoperability for IIoT systems. |
| Energy efficiency and fault tolerance for continuous operations (*Feng et al., 2023*; *Jin et al., 2023a*) | Minimise energy consumption and enhance fault tolerance to ensure dependable performance in harsh environments. | Optimize power efficiency with energy-aware DBA algorithms, implement adaptive sleep modes, and integrate fault-tolerant optical components. | Achieve 15–25% energy savings, ensure 99.999% network reliability, and maintain continuous operation in EMI-prone environments. |
| Deterministic capability and process automation (*ITU-T, 2024a*) | Provide bounded latency and predictable data transmission for industrial automation and process control. | Collaborate with ITU-T and other standardization bodies to define deterministic control protocols for IPONs and enhance predictability. | Achieve bounded latency, predictable performance, and synchronized communication for process automation and IIoT. |

manufacturers providing new capabilities. Still, this network also has some limitations, as listed below, and targets to achieve given in Table 9.

- **Traffic bursts and roaming delay**: FTTR applications inherently involve multiple access points, and users can also be mobile. This generates traffic bursts and increases roaming delay from one room to another. *Limitation*: These scenarios make maintaining consistently low latency and minimal roaming delays difficult, especially without precise traffic forecasting mechanisms (*Zang, Cao & Hong, 2024*).

- **Installation complexity and costs**: The deployment of FTTR systems requires skilled engineers for optical cable installation, and many residences lack adequate interior wiring to support such systems. *Limitation*: The installation process is time-consuming, averaging 30 min per room, and varies depending on the building's design and dimensions, contributing to high deployment costs (*Philpot & Wilson, 2022*).

- **Immature standardization**: Existing standardization efforts, such as ITU-T Q3/15 and ITU-T G.9940, encourage interoperability but remain insufficiently mature to support widespread industry adoption fully. *Limitation*: The lack of comprehensive and universally adopted standards limits the scalability and seamless integration of FTTR solutions across diverse markets and ecosystems (*Philpot & Wilson, 2022*).

## Industrial PON

The advancement of industrial networks is being propelled by industrial applications, which are leading to more automation, greater monitoring, and intelligent interconnection. Industrial Passive Optical Network (IPON) systems are being utilised to modernize and enhance current networks, facilitating the provision of high-speed

connections to fulfil essential demands. The ETSI GR-F5G-007 (*ETSI Recommendations, 2023*) outlines a universal network structure, as shown in Fig. 18, that utilizes PON to enable these applications and consolidate several access services onto a single platform. Figure 18 depicts the industrial PON architecture, wherein a central OLT interfaces with several ONUs dispersed around the manufacturing floor.

The PON facilitates high-bandwidth, low-latency communication, ensuring deterministic data transmission for IIoT devices, robotics, and time-sensitive applications. It consolidates field networks for instantaneous machine communication, office networks for operational oversight, and surveillance systems for high-definition video streams. The design utilizes time-sensitive networking (TSN) and QoS techniques to provide dependable performance, scalability for thousands of devices, and energy efficiency for continuous operations in challenging workplaces. Operators, vendors, and researchers are actively investigating the use cases of industrial PON with the following major goals/challenges given in Table 10.

# CONCLUSION

The performance of TDMA-PONs relies heavily on a DBA mechanism to improve bandwidth utilization. This article presented the technological advancement in the PON standards and discussed the working of the DBA engines in terms of frame structure for US and DS for the IEEE and ITU PONs. Higher data rates, traffic classification, error correction, security concerns, energy efficiency, multi-wavelength operations, and wavelength channel mobility for the IEEE PONs [EPON?10G-EPON?NG-EPON] and ITU-T PONs [A/BPON?GPON?XG-PON?TWDM PON] have all led to more complex frame structures (DBAs) over time. Following on, the DBAs proposed in the literature were evaluated based on performance indicators, *i.e.*, frame and channel idle time, QoS, and throughput. Looking ahead, as 6G networks promise to deliver dynamic and immersive applications that bridge the real and virtual worlds, the role of DBA for achieving 1 ms latency in NG-MFH networks will become even more significant (*Wong & Ruan, 2023*; *Fayad, Cinkler & Rak, 2024*) and focus on the following research directions.

## NG-MFH research directions

1. Abrupt traffic increases are difficult for current approaches to handle. Future research should investigate hybrid AI-heuristic methods for burst absorption in real time.
2. AI-driven DBA methods increase computational overhead and energy usage but boost efficiency. For scalability, distributed DBA techniques and lightweight learning models are required. In addition, energy-efficient DBA scheduling and Green AI methodologies should be focused on in the future.

New PON applications like FTTR and industrial PON require strict latency and bandwidth requirements, making design of DBAs an active research area with the following unresolved and actionable research directions:

### FTTR research directions

1. Develop algorithms for predictive and roaming-aware DBA to retain latency below 10 ms during traffic bursts and guarantee seamless roaming.
2. Investigate hybrid DBA methodologies that emphasize latency-sensitive applications, such as augmented reality/virtual reality and telemedicine, achieving less than 50 ms latency and 99% resource utilization.
3. Design energy-efficient DBA algorithms that engage power-saving modes during periods of low traffic, targeting a 15-20% reduction in energy consumption.
4. Standardize DBA protocols to achieve 90% device interoperability among vendors by 2026.

### Industrial PON research directions

1. Implement energy-efficient DBA algorithms and adaptive sleep modes to achieve a reduction in energy consumption of 15–25%, while maintaining 99.999% network reliability in challenging industrial settings.
2. Develop hierarchical DBA schemes to achieve 100% QoS compliance for IIoT devices, accommodating over 10,000 devices and ensuring seamless protocol interoperability.
3. Collaborate with standardization bodies to establish deterministic control protocols for IPONs, ensuring predictable performance and synchronized communication in process automation.

On a final note, PONs will continue to evolve and target to achieve 200 Gb/s. This would require discoveries and improvisations in the existing standard algorithms/architectures.

### Funding
This work was supported by the IRC for Communication Systems and Sensing, KFUPM, KSA, through project no. INCS2505. The funders had no role in study design, data collection and analysis, decision to publish, or preparation of the manuscript.

### Grant Disclosures
The following grant information was disclosed by the authors:
IRC for Communication Systems and Sensing, KFUPM, KSA: INCS2505.

### Competing Interests
The authors declare that they have no competing interests.

### Author Contributions
- Kamran Ali Memon conceived and designed the experiments, performed the experiments, analyzed the data, prepared figures and/or tables, authored or reviewed drafts of the article, and approved the final draft.

- Syed Saeed Jaffer conceived and designed the experiments, prepared figures and/or tables, and approved the final draft.
- Muhammad Ali Qureshi analyzed the data, prepared figures and/or tables, authored or reviewed drafts of the article, and approved the final draft.
- Khurram Karim Qureshi conceived and designed the experiments, performed the experiments, authored or reviewed drafts of the article, and approved the final draft.

## Data Availability

This is a literature review.

## Supplemental Information

Supplemental information for this article can be found online at http://dx.doi.org/10.7717/peerj-cs.2863#supplemental-information.

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
