# Peer review of "Dynamic bandwidth allocation in time division multiplexed passive optical networks: a dual-standard analysis of ITU-T and IEEE standard algorithms"

_PeerJ Computer Science, doi:10.7717/peerj-cs.2863_

## Round 0.1 · original submission · Minor Revisions

Be sure to address all comments from the reviewers, they are mostly minor.

Reviewer 1 ·

Basic reporting

1. The paper reviews DBA algorithms based on ITU-T and IEEE standards, and includes some discussions on related DBA works in emerging areas such as mobile fronthaul and industrial PON. An almost similar review of DBAs in EPON and GPON was published in 2020 by Thangappan, T., Therese, B., Suvarnamma, A., and Swapna, G. S. (2020). Review the dynamic bandwidth allocation of GPON and EPON. Journal of Electronic Science and Technology, 18(4):100044, which is also one of the references in this manuscript.
2. A more recent review of this topic is timely considering current trends that have shifted from traditional DBA to the incorporation of AI/ML, energy efficiency, and security.
3. The introduction section adequately introduces the subject. However, the second and third paragraphs of this section have significant textbook material, too basic to be discussed in an academic journal. They should be summarized to be more precise.

Experimental design

1. The organization of this manuscript is quite poor. The discussion on PON architectures is too lengthy and some parts are redundant. Figures 1, 5, 7 and 11 are based on similar concept of TDM-PON where main changes involve splitter ratio, fiber length and wavelengths with the different PON generations. One basic figure is sufficient with explanation on the major changes when network architecture upgrades.
2. Figure 1 and Table 2 should come first followed by explanations on the different EPON and GPON architectures, not the other way around.

Validity of the findings

1. Discussion on network architectures also can be shortened focusing on factors that affect DBA. Focus should be given on the aspects that have direct influence on DBA such as reporting, scheduling, service interval, frame structure (fields that important in DBA) etc. Critical analysis on how these aspects differ for different PON generations will be helpful.
2. The discussion on DBAs should also be assisted with several flow charts showing the DBA classification. It can be process-based such as in by Thangappan, T et al (2020) or objective-based (e.g. QoS, ML, Security, Energy efficiency etc). It is challenging to follow the discussion in its current state as the writing is more towards mere reporting than critical review analysis.
3. The conclusion identifies some future directions particularly in 6G, industrial PON. However, it is not comprehensive and fails to consider overall trends in DBA research e.g. AI/ML.

Additional comments

1. There are many formatting and language mistakes in the manuscript including tables and figures that requires professional proofreading service. Some abbreviations in the table not defined.
2. The figure qualities are quite poor, inconsistency in font size and type. Similar equipment like ONU, splitter have many representations in these figures.
3. Avoid use of informal phrases like "Thanks to"...
4. Explanation for Eq 1 need to be corrected as it contains parameters/symbols not in the equation.
5. Explanation for PON standardization on page 3 and 4 has duplicate content.
6. The in-text reference citation format needs to be improved e.g. follows APA. At least there should be brackets to distinguish between references and text.

Reviewer 2 ·

Basic reporting

The author/s has mainly focused more on the review part where lot of work is already carried out and less focus on methodology and practical implementations of the original work.

A few observations are:

1. Less focus on new approaches like 6G implementation compatibility, as it's only a part of the conclusion and left for future work, however, a few challenges can be discussed in the manuscript.
2. The authors need to mention a few traffic scenarios with the solutions and implementation.

Only minor changes are required to be done.

Experimental design

Mostly review part so not relevant.

Validity of the findings

Review.

Reviewer 3 ·

Basic reporting

The authors present an in-depth survey of all existing PON variants with a focus on DBA technologies. The paper touches also on novel technology trends, including FTTR, Industrial PON and VHSP.

Experimental design

The paper mainly copies information from other sources (papers, standards, etc.). Often, information is presented in a rather unconnected manner with plenty of words, often without balancing well enough different levels of abstraction. This means, very detailed information stands side-by-side high-level information without pointing out interconnections comprehensively enough.
It is written with a good level of English language. It seeks to give an overview of all PON technologies and its DBA mechanisms.

Validity of the findings

The quality of the paper is at best mediocre. If such a quality can be accepted for this journal, I recommend the following revisions:

General suggestions:
* * *
- Try to balance the scaling of figures. Some fonts in figures appear huge, others tiny

- Overall, the paper appears quite lengthy, often just dropping unconnected pieces of information, especially in section “ARCHITECTURAL AND TECHNOLOGICAL ADVANCEMENTS IN PONS”. A good survey should, however, give the reader background on how this information interrelates and it should put developments and facts into context for a wider understanding beyond the knowledge of details. A positive counterexample is Table 2, which gives context.

- The structure of the paper is hard to follow. Numbers for sections and (sub-)subsections should be added to the headers, such as “2.3.3 XXXXX”

- Whenever figures are copied from other sources, it must (!) be marked as such. Often this has not been done.


Specific suggestions:
* * *
- Line 42: Typo “dire”

- Line 65 “WDM-PONs are better than TDM-PONs because”; “better” is the wrong word here. They have pros and cons, as explained correctly.
Suggestion: “WDM-PONs provide a consistent service with reserved bandwidth (…)”

- Line 88 & 94 repetition: “IEEE develops Ethernet-based PON standards like EPON, ensuring interoperability and driving innovation through research and updates (…)”

- Line 99 ff. & Fig. 3: Not completely clear how NG-PON 1 and NG-PON 2 relates to the different standards – Suggestion: Name it in Fig. 3, not only in the text.

- Line 100: What does the abbreviation “ATM” stand for?

- Line 124: Where in Fig. 3 is this shown?

- Line 135: “packetsEffenberger”

- Line 184 ff.: The way these exploration topics are listed is not common for scientific journals. Some of the points are also strangely put:
o “Many aspirant researchers aim to understand both the architectural development and DBA structures through standard reports”
Why “aspirant”? The kind of researchers is irrelevant.
o “ET4: What are the open research issues for the new PON applications such as FTTR and industrial PONs?
Ans: Yes, section 6 focuses on this question.”
Here, what does “Yes” refer to?
Suggestion: Formulate the scope of the paper not in full text, rather than Q&A points

- Fig. 4 Typo “Netwroks”

- Line 240: “spilt ratio”

- Multiple times in the paper: “Comprises of” without “of”

- Line 325: “Table reftab:comparison-pon-standards”

- Some texts in Table 2 are misaligned

- Tables 4 and 6 are misaligned

- Line 723: ” radio remote head” (RRH) => remote radio head

- Line 722 ff. Leaving only antennas at the RRH is just one of many different fronthaul split point options.

- Line 811: There are FTTR standardization efforts: ITU-T Q3/15 and ITU-T G.9940

---

## Round 0.2 · Major Revisions

Please fully address these additional comments

Reviewer 1 ·

Basic reporting

The manuscript demonstrates a clear structure and adheres to PeerJ’s formatting guidelines. The authors have made noticeable improvements in language and formatting since the initial submission, which enhances readability. The detailed discussions on ITU-T and IEEE standards and the evolution of PON architectures provide valuable background for readers. However, these sections could be condensed to allocate more space for deeper analysis of DBA algorithms, which is central to the paper's objectives.

Additionally, PeerJ’s open-access model ensures that this work will be widely accessible to researchers across multiple disciplines. Strengthening the critical analysis and synthesis of DBA research will make this manuscript a key reference in the field and fully leverage the journal’s broad reach and visibility.

Experimental design

The manuscript aims to review DBA algorithms in different generations of ITU-T and IEEE PONs comprehensively. While the background on standards and architectures is thorough, the critical evaluation of DBA algorithms is still lacking. A review paper’s primary purpose is not only to summarize but also to synthesize and critique prior research, highlighting trends, challenges, and opportunities.

The authors note limitations in page numbers, which is understandable. However, reprioritizing content by condensing descriptive background sections and focusing on trade-offs, assumptions, and applicability of DBA algorithms could provide room for this critical analysis without exceeding page limits. This approach will not only enhance the manuscript’s relevance but also align with PeerJ’s goal of publishing high-quality, impactful work.

Validity of the findings

The manuscript references relevant prior work and provides a comprehensive overview of DBA mechanisms. However, the analysis remains largely descriptive. To validate the findings as a review paper, the manuscript should:

1. Critically compare DBA algorithms, discussing strengths, weaknesses, and practical applicability to next-generation applications like 5G/6G fronthaul and Industrial PONs.
2. Provide actionable insights for future research, such as specific challenges (e.g., computational overheads in AI/ML-based DBA or latency concerns in low-delay environments).

PeerJ’s platform allows this work to reach a broad audience, making it essential to include critical evaluations that add value to the field and inform both researchers and practitioners.

Additional comments

Thank you for your significant effort in revising the manuscript. While improvements in language and formatting are noted, I strongly encourage you to address the critical analysis issues highlighted earlier. This will significantly enhance the quality and impact of your paper, aligning with PeerJ’s high standards and broad reach.

Key Suggestions for Improvement:
1. Reprioritize Content: Condense detailed background sections to allocate more space for critical discussions of DBA mechanisms.
2. Enhance Critical Depth: Discuss trade-offs, applicability, and challenges of existing DBA algorithms, supported by quantitative examples or insights from the cited works.
3. Improve Future Directions: Provide specific, actionable challenges and opportunities for DBA in emerging applications like 5G/6G fronthaul, FTTR, and Industrial PONs.

Publishing in PeerJ provides a unique opportunity to make your work accessible to a global audience while contributing to an interdisciplinary understanding of DBA. I appreciate your efforts and encourage you to leverage this opportunity by addressing these critical aspects.

---

## Round 0.3 · Minor Revisions

Please respond to these last minor comments from the reviewer

Reviewer 1 ·

Basic reporting

1. Is the review of broad and cross-disciplinary interest and within the scope of the journal?
The paper’s focus on dynamic bandwidth allocation (DBA) in passive optical networks is relevant not only to optical communication researchers but also to broader networking and telecommunications audiences. By addressing both IEEE and ITU-T standards, the review crosses the traditional boundaries of each standard’s research community. This dual-standard perspective can appeal to a wide set of stakeholders, including those working on network architecture, 5G/6G infrastructure, and industrial automation. It thus fits well within a journal scope that encompasses networking, communication technologies, and their applications.

2. Has the field been reviewed recently? If so, is there a good reason for this review (different point of view, accessible to a different audience, etc.)?
Although multiple reviews on PON technologies and DBA algorithms have been published in the past, this article offers a fresh angle by explicitly comparing and contrasting the two major standardization streams (IEEE vs. ITU-T). The inclusion of next-generation PON applications e.g. 5G fronthaul, FTTR, and industrial PON further justifies the need for an updated review, given the rapid technological advancements and the expanding scope of optical access networks. This timely perspective is beneficial for both newcomers and experts looking to understand the latest trends and open challenges in DBA research.

3. Does the Introduction adequately introduce the subject and make it clear who the audience is/what the motivation is?
The Introduction gives sufficient background on the evolution of PON technologies, emphasizing why bandwidth allocation is crucial and how DBA mechanisms have evolved to meet growing network demands. The motivation bridging the gap between different standards and highlighting new use cases comes across clearly. The target audience appears to be researchers, industry practitioners, and graduate students who seek an in-depth technical overview of DBA algorithms and their real-world implications. However, some simplification of technical jargon and more explicit statements of the paper’s unique contributions early on could make the motivation even clearer to a broader readership.

Experimental design

1. Is the Survey Methodology consistent with a comprehensive, unbiased coverage of the subject? If not, what is missing?
Overall, the paper’s methodology appears to be systematic, covering a wide range of relevant DBA algorithms under both IEEE and ITU-T standards. The authors discuss foundational algorithms as well as newer techniques and applications, which contributes to a balanced viewpoint.

2. Are sources adequately cited? Quoted or paraphrased as appropriate?
The paper cites a variety of well-known journals and conference proceedings, indicating that the authors have drawn from reputable sources. In most places, the references appear in line with standard scholarly practices, with direct quotations kept to a minimum and proper paraphrasing in place.

3. Is the review organized logically into coherent paragraphs/subsections?
The manuscript is divided into clear sections, such as an introduction, background on PON standards, DBA algorithms for different generations of PON, and potential applications. This structure helps readers navigate from foundational concepts to more advanced topics.

Validity of the findings

1. Is there a well developed and supported argument that meets the goals set out in the Introduction?
Yes, the paper’s core argument that dynamic bandwidth allocation in passive optical networks needs to be revisited in light of both IEEE and ITU-T standards comes across strongly. The authors systematically discuss how each standard addresses upstream bandwidth allocation, linking this back to the goals stated in the Introduction: namely, to compare multiple PON generations and highlight open challenges in modern optical access networks.

2. Does the Conclusion identify unresolved questions / gaps / future directions?
The Conclusion does mention emerging applications (e.g., 5G fronthaul, Industrial PON, Fiber-to-the-Room) and the associated need for more sophisticated DBA mechanisms. However, the paper could be even more explicit in laying out concrete research gaps and future directions. For instance, highlighting open issues such as the challenges of low-latency DBA under bursty traffic or the trade-offs between complexity and scalability in hybrid PON architectures would give readers clearer guidance on where additional work is most needed. Overall, adding a concise bullet-point list or short subsection on “Future Research Directions” could make these unresolved questions more visible and actionable.

Additional comments

Additional Comments

The authors have significantly improved the manuscript by addressing the crucial comments, particularly regarding the criticalness of the review. Only a few minor points remain to be addressed, as follows:

Proofreading and Language: Although the manuscript reads well overall, there are still occasional typographical and spelling errors that could distract readers. A thorough proofreading possibly with assistance from a professional editing service or a native English speaker would help ensure the writing meets the highest academic standards.

Figure Quality and Consistency: Several figures have inconsistent font sizes, styles, and labeling conventions. Standardizing these elements would make the visual presentation clearer and more professional. High-resolution images and uniform design (e.g., using the same font type and size for all labels and legends) will enhance readability and aesthetic coherence across the paper.

Clarity in Data Presentation: When including tables or graphs, ensure that all axes, headers, and units are clearly labeled. For complex tables, consider providing short explanatory notes or captions to guide readers through the data.

Section Flow and Subheadings: While the organization is logical, adding subheadings especially in long sections can help readers navigate the content more easily. Shorter paragraphs and the use of bullet points for key findings or recommendations could further improve clarity.

Consistency in Terminology and Acronyms: Double-check that acronyms (e.g., DBA, PON, ONU) are spelled out at first mention and used consistently throughout. Inconsistencies in how acronyms are introduced or capitalized can cause confusion, especially for readers new to the subject.

Future Research Directions: Although the Conclusion touches on emerging applications and the need for more advanced DBA schemes, clearly identifying unresolved issues, open questions, or specific research challenges would strengthen the paper’s impact and guide future work in this area.

---

## Round 0.4 · accepted · Accept

The manuscript has addressed reviewer concerns and requests for changes.